# Remodeling of skeletal muscle myosin metabolic states in hibernating mammals

Christopher TA Lewis[1]*, Elise G Melhedegaard[1], Marija M Ognjanovic[1], Mathilde S Olsen[1], Jenni Laitila[1], Robert AE Seaborne[1,2], Magnus Gronset[3], Changxin Zhang[4], Hiroyuki Iwamoto[5], Anthony L Hessel[6,7], Michel N Kuehn[6,7], Carla Merino[8], Nuria Amigo[8], Ole Frobert[9,10], Sylvain Giroud[11,12], James F Staples[13], Anna V Goropashnaya[14], Vadim B Fedorov[14], Brian Barnes[14], Oivind Toien[14], Kelly Drew[14], Ryan J Sprenger[15], Julien Ochala[1]*

[1]Department of Biomedical Sciences, University of Copenhagen, Copenhagen, Denmark; [2]Centre for Human and Applied Physiological Sciences, Faculty of Life Sciences & Medicine, King's College London, London, United Kingdom; [3]Department of Cellular and Molecular Medicine, University of Copenhagen, Copenhagen, Denmark; [4]Department of Computational Medicine and Bioinformatics, University of Michigan, Ann Arbor, United States; [5]Spring-8, Japan Synchrotron Radiation Research Institute, Hyogo, Japan; [6]Institute of Physiology II, University of Muenster, Muenster, Germany; [7]Accelerated Muscle Biotechnologies Consultants, Boston, United States; [8]Biosfer Teslab, Reus, Spain; [9]Department of Clinical Medicine, Faculty of Health, Aarhus University, Aarhus, Denmark; [10]Faculty of Health, Department of Cardiology, Örebro University, Örebro, Sweden; [11]Energetics Lab, Department of Biology, Northern Michigan University, Marquette, United States; [12]Research Institute of Wildlife Ecology, Department of Interdisciplinary Life Sciences, University of Veterinary Medicine Vienna, Vienna, Austria; [13]Department of Biology, University of Western Ontario, London, Canada; [14]Center for Transformative Research in Metabolism, Institute of Arctic Biology, University of Alaska Fairbanks, Fairbanks, United States; [15]Department of Zoology, University of British Columbia, Vancouver, Canada

*For correspondence:
christopher.lewis@sund.ku.dk
(CTAL);
julien.ochala@sund.ku.dk (JO)

**Abstract** Hibernation is a period of metabolic suppression utilized by many small and large mammal species to survive during winter periods. As the underlying cellular and molecular mechanisms remain incompletely understood, our study aimed to determine whether skeletal muscle myosin and its metabolic efficiency undergo alterations during hibernation to optimize energy utilization. We isolated muscle fibers from small hibernators, *Ictidomys tridecemlineatus* and *Eliomys quercinus* and larger hibernators, *Ursus arctos* and *Ursus americanus*. We then conducted loaded Mant-ATP chase experiments alongside X-ray diffraction to measure resting myosin dynamics and its ATP demand. In parallel, we performed multiple proteomics analyses. Our results showed a preservation of myosin structure in *U. arctos* and *U. americanus* during hibernation, whilst in *I. tridecemlineatus* and *E. quercinus*, changes in myosin metabolic states during torpor unexpectedly led to higher levels in energy expenditure of type II, fast-twitch muscle fibers at ambient lab temperatures (20 °C). Upon repeating loaded Mant-ATP chase experiments at 8 °C (near the body temperature of torpid animals), we found that myosin ATP consumption in type II muscle fibers was reduced by 77–107% during torpor compared to active periods. Additionally, we observed Myh2 hyperphosphorylation during torpor in *I. tridecemilineatus*, which was predicted to stabilize the myosin molecule. This may act as a potential molecular mechanism mitigating myosin-associated increases in skeletal muscle energy expenditure during periods of torpor in response to cold exposure.

Altogether, we demonstrate that resting myosin is altered in hibernating mammals, contributing to significant changes to the ATP consumption of skeletal muscle. Additionally, we observe that it is further altered in response to cold exposure and highlight myosin as a potentially contributor to skeletal muscle non-shivering thermogenesis.

## eLife assessment

The work by Lewis and co-workers presents **important** findings on the role of myosin structure/energetics on the molecular mechanisms of hibernation by comparing muscle samples from small and large hibernating mammals. The **solid** methodological approaches have revealed insights into the mechanisms of non-shivering thermogenesis and energy expenditure.

## Introduction

Hibernation is an adaptive strategy employed by many animals aiming to decrease their metabolic rate and improve survival, particularly during harsh, winter conditions where food supply is limited, and thermogenic demands are high (*Geiser, 2013*). During hibernation, mammals typically undergo a decrease in body temperature, heart, and breathing rates (*Jani et al., 2013*; *Milsom and Jackson, 2024*; *Sprenger and Milsom, 2022*). In so-called fat-storing hibernators, this is inherently accompanied by prolonged fasting and fatty acids become the main substrate for energy provision (*Florant, 1998*; *Giroud et al., 2020*). Besides these common features associated with overall metabolic depression, there are also significant inter-species differences in the underlying strategies. For instance, small (<8 kg) fat-storing hibernating mammals such as 13-lined ground squirrels (*Ictidomys tridecemlineatus*) or garden dormice (*Eliomys quercinus*) experience extended bouts of a hypo-metabolic state (torpor), punctuated by spontaneous periods of interbout euthermic arousals (IBA), during which metabolic activity transiently increases back to basal levels. During torpor, metabolic rate decreases below 5% of euthermic values and core body temperatures decrease from 35°C–38°C to 4°C–8°C (*Sprenger et al., 2018*; *Haugg et al., 2024*). In contrast, either medium (10–20 kg, e.g. European badger, *Meles meles*) or large (>20 kg, e.g. Eurasian brown bear, *Ursus arctos*, and American black bear, *Ursus americanus*) hibernating mammals exhibit a pronounced hypo-metabolic state (as low as 25% of their basal metabolic rate in the case of bears), but only experience a mild decline in body temperature (to 32–35°C depending on body size) that lasts for several winter months (*Geiser, 2013*; *Tøien et al., 2011*; *Evans et al., 2023*). While species-specific physiological patterns are well-documented, the molecular and cellular mechanisms operative in individual organs to achieve these remain largely undefined.

Skeletal muscle constitutes approximately 45–55% of body mass and serves as a major determinant of basal metabolic rate and heat production (*Zurlo et al., 1990*; *Sylow et al., 2021*). Previous studies have uncovered some specific metabolic changes in skeletal muscle during hibernation (*Giroud et al., 2020*). For example, a decrease in mitochondrial respiration, as well as a suppression of ATP production capacity, has previously been documented in skeletal muscles from *I. tridecemlineatus* during torpor (*James et al., 2013*). Previous work has demonstrated that ground squirrels require an optimal dietary ratio of monounsaturated to polyunsaturated fats, approximately 2:1, during their fat-storing period to facilitate effective hibernation (*Frank and Storey, 1995*). The proteome of *I. tridecemlineatus* has then been shown to be enriched for fatty acid β-oxidation during periods of torpor. However, some reliance upon carbohydrate metabolism is maintained. The activity of phosphoglucomutase (PGM1) is even increased during torpor (*Hindle et al., 2011*). In *U. arctos*, skeletal muscle exhibits a transition from carbohydrate utilization to lipid metabolism, coupled with a reduction in whole-tissue ATP turnover (*Chazarin et al., 2019*).

Muscle is organized into an array of fibers containing repeating sarcomeres, which are crucial for regulating not only contraction but also metabolism and thermogenesis (*Gordon et al., 2000*). Until recently, energy consumption in skeletal muscle was thought to be primarily linked to the activity of myosin during muscular contraction (*Gordon et al., 2000*). Additionally, thermogenesis in skeletal muscle was previously attributed primarily to the electron transport system, in some cases link to uncoupling of sarcoplasmic reticulum calcium ATPase (SERCA) (*Periasamy et al., 2017*; *González-Alonso et al., 2000*). However, growing evidence that the sarcomeric metabolic rate and thermogenesis are also controlled by 'relaxed' myosin molecules is emerging (*McNamara et al., 2015*).

**eLife digest** Many animals use hibernation as a tactic to survive harsh winters. During this dormant, inactive state, animals reduce or limit body processes, such as heart rate and body temperature, to minimise their energy use. To conserve energy during hibernation, animals can use different approaches. For example, garden dormice undergo periodic states of extremely low core temperatures (down to 4–8°C); whereas Eurasian brown bears see milder temperature drops (down to 23–25°C).

An important organ that changes during hibernation is skeletal muscle. Skeletal muscle typically uses large amounts of energy, making up around 50% of body mass. To survive, hibernating animals must change how their skeletal muscle uses energy. Traditionally, active myosin – a protein found in muscles that helps muscles to contract – was thought to be responsible for most of the energy use by skeletal muscle. But, more recently, resting myosin has also been found to use energy when muscles are relaxed. Lewis et al. studied myosin and skeletal muscle energy use changes during hibernation and whether they could impact the metabolism of hibernating animals.

Lewis et al. assessed myosin changes in muscle samples from squirrels, dormice and bears during hibernation and during activity. Experiments showed changes in resting myosin in squirrels and dormice (whose temperature drops to 4–8°C during hibernation) but not in bears. Further analysis revealed that cooling samples from non-hibernating muscle to 4–8°C increased energy use in resting myosin, thereby generating heat. However, no increase in energy use was found after cooling hibernating muscle samples to 4–8°C. This suggest that resting myosin generates heat at cool temperatures – a mechanism that is switched off in hibernating animals to allow them to cool their body temperature.

These findings reveal key insights into how animals conserve energy during hibernation. In addition, the results show that myosin regulates energy use in skeletal muscles, which indicates myosin may be a potential drug target in metabolic diseases, such as obesity.

Myosin heads in passive muscle (pCa >8), can be in different resting metabolic states that maintain a basal level of ATP consumption. In the 'disordered-relaxed' (DRX) state, myosin heads are generally not bound to actin and structurally exist in a conformation (so-called ON state) where they primarily exist freely within the interfilamentous space in the sarcomere (*Stewart et al., 2010*; *Grinzato et al., 2023*). In the 'super-relaxed' (SRX) state, myosin heads adopt a structural conformation against the thick filament backbone (so-called OFF state; *Hooijman et al., 2011*). This conformation sterically inhibits the ATPase site on the myosin head, significantly reducing both ATP turnover in these molecules and, therefore heat production. The SRX state has an ATP turnover rate five to ten times lower than that of myosin heads in the DRX state (*Cooke, 2011*). A 20% shift of myosin heads from SRX to DRX is predicted to increase whole-body energy expenditure by 16% and double skeletal muscle thermogenesis (*Cooke, 2011*). What remains to be determined is whether, across mammals, increasing the proportions of myosin in the DRX or SRX states serves as a physiological molecular mechanism to fine-tune metabolic demands and thermogenesis. This includes potential contributions to whole-body metabolic depression observed during hibernation.

Hence, in the present study, we hypothesized that a remodeling of the proportions of myosin DRX and SRX conformations within skeletal muscles occurs and is a major suppressor of ATP/metabolic demand during hibernation. A recent study on *I. tridecemlineatus* cardiac muscle supports this hypothesis, finding higher proportions of SRX during torpor (*Toepfer et al., 2020*). We examined isolated skeletal myofibers extracted from both small and large hibernating mammals - *I. tridecemlineatus*, *E. quercinus*, *U. arctos,* and *U. americanus*. We employed a multifaceted approach: loaded Mant-ATP chase experiments to assess myosin conformation and ATP turnover time, X-ray diffraction for sarcomere structure evaluation, and proteomic analyses to quantify differential PTMs.

## Results

### Resting myosin metabolic states are preserved in skeletal muscles fibers of hibernating *Ursus arctos* and *Ursus americanus*

To investigate whether resting myosin DRX and SRX states and their respective ATP consumption rates were altered during hibernation, we utilized the loaded Mant-ATP chase assay in isolated permeabilized muscle fibers from *U. arctos* obtained during either summer (active period) or winter (hibernating period). A total of 104 myofibers were tested at ambient lab temperatures (20 °C). A representative decay of the Mant-ATP fluorescence in single muscle fibers is shown, indicating ATP consumption by myosin heads (*Figure 1A*). In both type I (myosin heavy chain - MyHC-I) and type II (MyHC-II) myofibers, we did not observe any change in the percentage of myosin heads in either the DRX (P1 in *Figure 1B*) or SRX states (P2 in *Figure 1C*). We also did not find any difference in their ATP turnover times as demonstrated by the preserved T1 (*Figure 1D*) and T2 values (*Figure 1E*). We calculated the ATP consumed by myosin molecules by using the following equation based on the assumption that the concentration of myosin heads within single muscle fibers is 220 µM (*Cooke, 2011*):

$$\text{Myosin ATP consumption (fiber} \times \text{min)} = (\text{P1/100}) \times 220 \times (60/\text{T1}) + (\text{P2/100}) \times 220 \times (60/\text{T2})$$

the ATP consumed by myosin molecules was not different between groups (*Figure 1F*). We repeated these experiments on 95 myofibers obtained from summer and winter from *U. americanus*. A representative decay of the Mant-ATP fluorescence in single muscle fibers is shown (*Figure 1G*). We found very similar results to the *U.* arctos that in both type I and type II muscle fibers, with no changes to resting myosin conformation, ATP turnover time or ATP consumption per fiber. These data indicate that myosin metabolic states are unchanged during hibernation in both *U. arctos* and *U. americanus*.

### Relaxed myosin metabolic states are disrupted in skeletal myofibers of small hibernators: *Ictidomys tridecemlineatus* and *Eliomys quercinus*

In smaller hibernators we utilized samples obtained from *E. quercinus* and *I. tridecemlineatus* during the summer active state (SA), IBA and torpor. In *E. quercinus*, a total of 146 myofibers were assessed at ambient lab temperatures. Consistent with *U. arctos* and *U. americanus*, we did not see any difference in the percentage of myosin heads in either the DRX (*Figure 2A*) or SRX states (*Figure 2B*). However, the ATP turnover time of myosin molecules in the DRX conformation (DRX T1) was 35% lower in IBA and torpor compared with SA in type I fibers and was 36% and 31% lower during IBA and torpor, respectively, compared to SA in type II fibers (*Figure 2C*). The ATP turnover time of myosin heads in the SRX state (SRX T2) was 28% lower in in both IBA and torpor compared to SA for type I fibers and was lower in torpor by 26% compared with TA for type II fibers (*Figure 2D*). All these changes were accompanied by a 56% greater myosin-based ATP consumption of in IBA compared to SA for type I fibers. In type II myofibers ATP consumption was 55% and 47% greater in IBA and torpor, respectively, compared with SA (*Figure 2E*). In *I. tridecemlineatus*, a total of 156 muscle fibers were evaluated. In accordance with *E. quercinus,* we did not observe any modification in the percentage of myosin heads in the DRX conformation (P1) in the SRX conformation (P2) in either fiber type between SA, IBA or torpor (*Figure 2F and G*). Nevertheless, DRX T1 was 35% lower for type I fibers in torpor compared to SA and was 29% lower in IBA and 46% in torpor compared to SA for type II fibers (*Figure 2H*). SRX T2 was significantly lower in both IBA, and torpor compared to SA for type I by 49% and 29%, respectively (*Figure 2I*). SRX T2 was also significantly lower in type II fibers in both IBA, and torpor compared to SA by 31% and 27%, respectively. Myosin-specific ATP consumption in type II muscle fibers during torpor was significantly higher at 99% compared to SA (*Figure 2J*).

To gain insights into the mechanisms that cause such unexpected myosin metabolic adaptive changes, we investigated whether these latter changes were accompanied by a structural alteration of myosin molecules. We collected and analysed X-ray diffraction patterns of thin muscle strips in *I. tridecemlineatus*. The ratio of intensities between the 1,0 and 1,1 reflections ($I_{1,1} / I_{1,0}$; *Figure 3A*) provides a quantification of myosin mass movement between thick and thin filaments. An increase in $I_{1,1} / I_{1,0}$ reflects myosin head movement from thick to thin filaments, where increasing $I_{1,1} / I_{1,0}$ tracks myosin head movement from thick to thin filaments, signaling a transition from the OFF to ON state (*Ma and Irving, 2022*). 1,0 and 1,1 equatorial intensities were quantified and the intensity ratio ($I_{1,1}$ to $I_{1,0}$) calculated. This intensity ratio was significantly lower in torpor compared to IBA (*Figure 3B*),

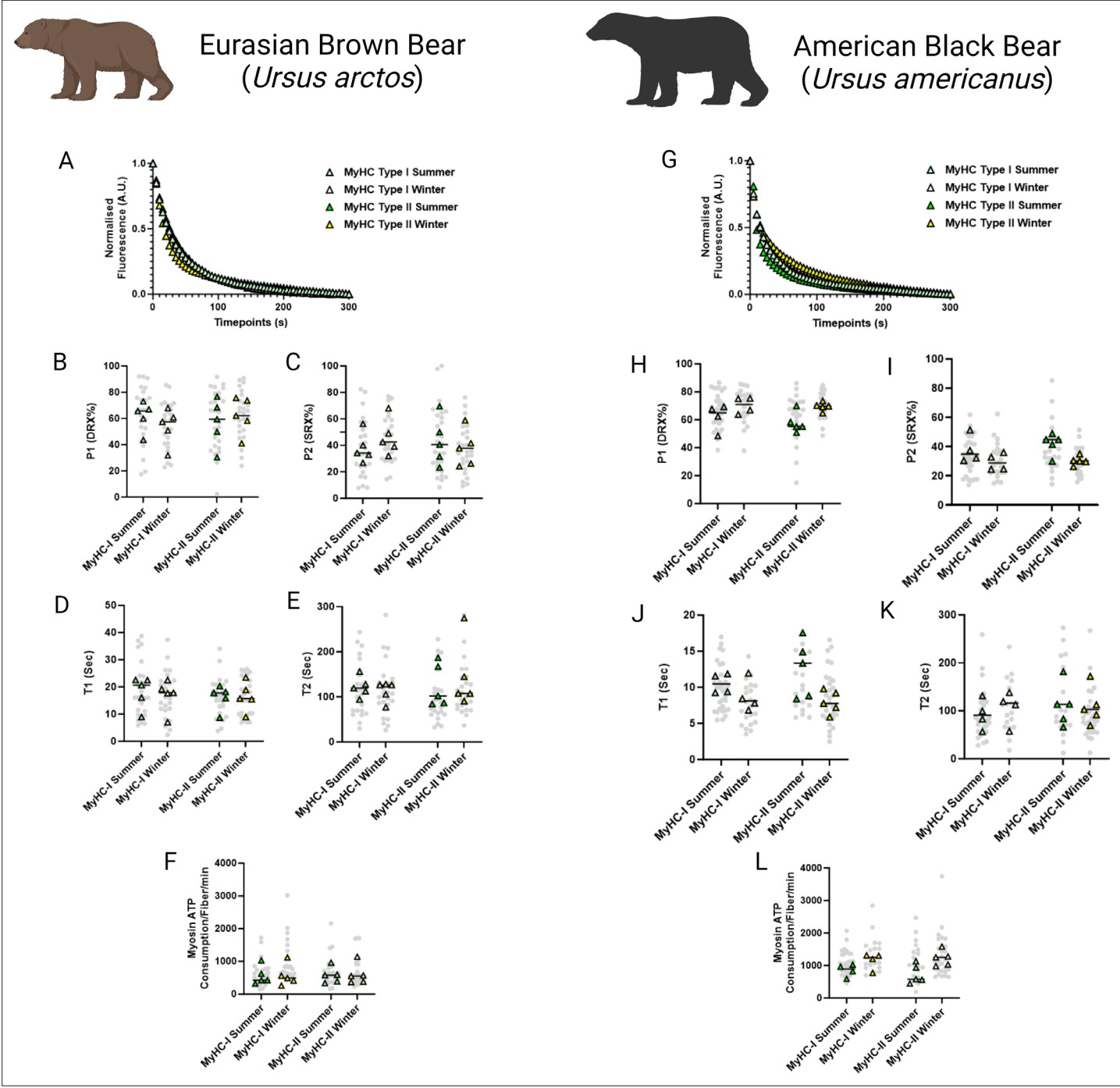

**Figure 1.** Myosin dynamics and myosin ATP consumption is unchanged in *Ursus arctos* and *Ursus americanus* during hibernation. (**A**) Representative fluorescence mant-ATP decays from single muscle fibers isolated from *Ursus arctos* skeletal muscle measured over 300 s. (**B–C**) Percentage of myosin heads in the P1/DRX (**B**) or P2/SRX (**C**) from *Ursus arctos* single muscle fibers obtained during summer (active) or winter (hibernating) periods. Values were separated based on each individual fiber was MyHC type I or MyHC type II. (**D**) T1 value in seconds denoting the ATP turnover lifetime of the DRX. (**E**) T2 value in seconds denoting the ATP turnover lifetime in seconds of the SRX. (**F**) Calculated myosin ATP consumption values of each single muscle fiber per minute. This was calculated using the equation shown in the Materials and methods section. (**G**) Representative fluorescence mant-ATP decays from single muscle fibers isolated from *Ursus americanus* skeletal muscle measured over 300 s. (**H–I**) Percentage of myosin heads in the P1/DRX (**G**) or P2/SRX (**H**) from *Ursus americanus* single muscle fibers obtained during summer (active) or winter (hibernating) periods. Values were separated based on each individual fiber was MyHC type I or MyHC type II. (**J**) T1 value in seconds denoting the ATP turnover lifetime of the DRX. (**K**) T2 value in seconds denoting the ATP turnover lifetime in seconds of the SRX. (**L**) Calculated myosin ATP consumption values of each single muscle fiber per minute. Grey circles represent the values from each individual muscle fiber which was analyzed. Colored triangles represent the mean value from an individual animal,

*Figure 1 continued on next page*

*Figure 1 continued*

8–12 fibers analyzed per animal. Statistical analysis was performed upon mean values. One-way ANOVA was used for statistical testing. n=5 individual animals per group. Figure created using BioRender.com and published using a CC BY-NC-ND license with permission.

suggesting more myosin heads are structurally OFF in torpor vs IBA. A reorientation of the myosin heads between OFF and ON states can also be captured by the M3 reflection along the meridional axis of diffraction patterns (*Figure 3A*). The M3 spacing represents the average distance between myosin crowns along the thick filament. An increase in M3 spacing signifies a reorientation of myosin heads from the OFF towards the ON states (*Ma et al., 2018a*). No differences were seen in M3 spacing or intensity (*Figure 3C and D*), indicating that the orientation of myosin crowns along the thick filament are similar across all conditions (SA, IBA and torpor). Thick filament length, measured here by the spacing of the M6 reflection (*Figure 3A*), was significantly greater in IBA compared to SA, and during torpor compared to both SA and IBA (*Figure 3E*). Taken together we report a unique structural signature of muscle during hibernating periods in *I. tridecemlineatus.*

## Myosin temperature sensitivity is lost in relaxed skeletal muscles fibers of hibernating *Ictidomys tridecemlineatus*

To mimic the drastic body temperature decrease experienced by small hibernators during torpor, we repeated the loaded Mant-ATP chase experiments in *I. tridecemlineatus* at 8 °C. A total of 138 myofibers were assessed and similar to when measured at ambient lab temperatures, no changes to the conformation of myosin states were observed (*Figure 4—figure supplement 1*). At these temperatures, we observed that during torpor the DRX T1 was 77% compared to SA and 107% higher compared to IBA in type II muscle fibers (*Figure 4A*). We further observed that during torpor the SRX T2 was 60% higher compared to SA and 64% higher compared to IBA (*Figure 4B*). We then calculated the 20°C to 8°C degree ratios, allowing us to define myosin temperature sensitivity of DRX T1, SRX T2 and myosin ATP consumption (*Figure 4C, D and E*). In SA and IBA, lowering the temperature led to a reduction in the myosin ATP turnover time of both the DRX and SRX, thus decreasing ATP consumption especially for type II muscle fibers. In contrast, during torpor lowering the temperature had opposite effects (*Figure 4C, D and E*). This observation was particularly prominent in type II muscle fibers (*Figure 4C and D*). From these results, we suggest that *I. tridecemlineatus* reduce their resting myosin ATP turnover rates in response to cold exposure to increase heat production via ATP hydrolysis.

## Hyper-phosphorylation of Myh2 predictably stabilizes myosin backbone in hibernating *Ictidomys tridecemlineatus*

Based on our findings of discrepancies between the dynamics of myosin at different temperatures, we wanted to obtain a greater understanding of changes at the protein level during these different hibernating states. The myosin molecule is well-known to be heavily regulated by post-translational modifications (PTMs; *Nag et al., 2017*; *Spudich, 2015*; *Bódi et al., 2021*; *Duggal et al., 2014*; *Huang et al., 2015*; *Vandenboom et al., 2013*; *Papadaki et al., 2022*). We assessed whether hibernation impacts the level of phosphorylation and acetylation on the Myh7 and Myh2 proteins from *I. tridecemlineatus*. Myh2 exhibited significant differences in phosphorylation sites during torpor compared to SA and IBA (*Figure 5A*). Three specific residues had significantly greater levels of phosphorylation: threonine 1039 (Thr1039-P), serine 1240 (Ser1240-P), and serine 1300 (Ser1300-P). These PTMs lie within the coiled-coil region of the myosin filament backbone (*Figure 5B and C*). To define whether they have functional implications during hibernation, we utilized EvoEF, an in silico programme which can characterize the effects of single amino acid residue substitutions on protein stability (*Huang et al., 2020*). For our analysis, we replaced the three native residues where PTMs were found by aspartic acid (Asp), which chemically resembles phosphothreonine (Thr-P) and phosphoserine (Ser-P) (*Haase et al., 2004*). Thr1039Asp, Ser1240Asp, and Ser1300Asp had higher protein stabilities compared to Thr1039, Ser1240, and Ser1300, as attested by $\Delta\Delta G_{Stability}$ values greater than zero (*Figure 5D*). The combination of these modifications does not counteract one another and thus are predicted to provide a high change in Myh2 stability ($\Delta\Delta G_{Stability}$ of 2.54, *Figure 5D*).

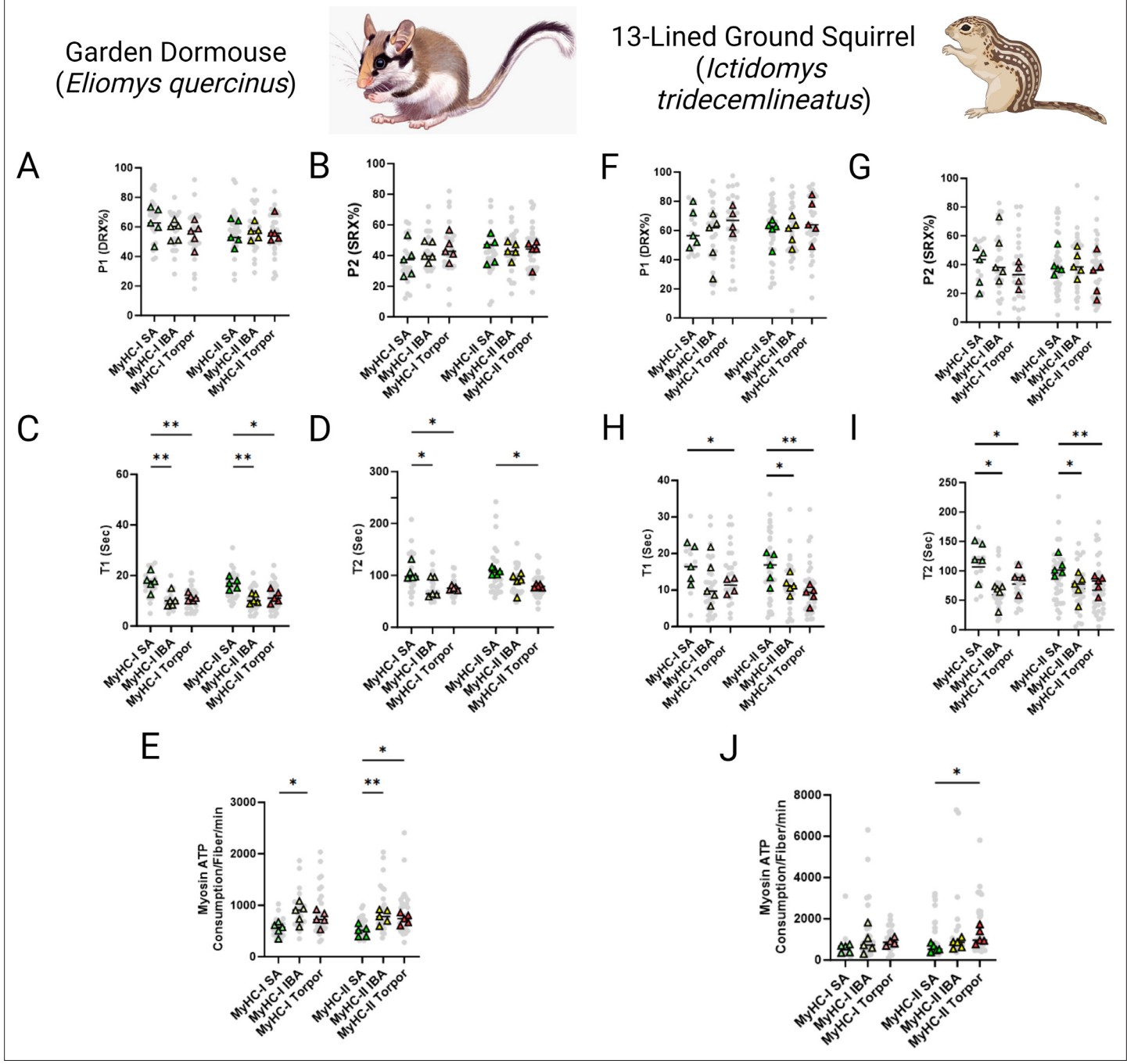

**Figure 2.** Myosin ATP turnover lifetime is reduced during hibernation in small hibernators, *Eliomys quercinus* and *Ictidomys tridecemlineatus*, resulting in an increase in myosin ATP consumption at ambient temperatures. (**A–B**) Percentage of myosin heads in the P1/DRX (**A**) or P2/SRX (**B**) from *E. quercinus* single muscle fibers obtained during active, interbout arousal (IBA) or torpor periods. Values were separated based on each individual fiber was MyHC type I or MyHC type II. (**C**) T1 value in seconds denoting the ATP turnover lifetime of the DRX in *E. quercinus*. (**D**) T2 value in seconds denoting the ATP turnover lifetime in seconds of the SRX in *E. quercinus*. (**E**) Calculated myosin ATP consumption values of each single muscle fiber per minute in *E. quercinus*. This was calculated using the equation shown in the Materials and methods section. (**F–G**) Percentage of myosin heads in the P1/DRX (**F**) or P2/SRX (**G**) from *I. tridecemlineatus* single muscle fibers obtained during summer active (SA), interbout arousal (IBA) or torpor periods. (**H**) T1 value in seconds denoting the ATP turnover lifetime of the DRX in *I. tridecemlineatus*. (**I**) T2 value in seconds denoting the ATP turnover lifetime in seconds of the SRX in *I. tridecemlineatus*. (**J**) Calculated myosin ATP consumption values of each single muscle fiber per minute in *I. tridecemlineatus*. Grey circles represent the values from each individual muscle fiber which was analyzed. Colored triangles represent the mean value from an individual animal, 8–12 fibers analyzed per animal. Statistical analysis was performed upon mean values. One-way ANOVA was used to calculate statistical

*Figure 2 continued on next page*

We did not identify any hibernation-related changes in Myh7 phosphorylation or acetylation, nor did we detect any alterations in Myh2 acetylation in *I. tridecemlineatus*.

To validate the significance of the PTM findings, a parallel PTM analysis was performed for *U. arctos*, a species in which myosin metabolic states are unaffected by hibernation (*Figure 1*). In this context, minimal changes in phosphorylated or acetylated residues were observed in the Myh7 and Myh2 proteins (*Figure 5—figure supplements 1 and 2*). These minimal modifications occurred on amino acids distinct from those in *I. tridecemlineatus* (*Figure 5—figure supplements 1 and 2*). Overall, our PTM analyses and related simulations indicate that torpor is associated with a Myh2 hyper-phosphorylation possibly impacting myosin filament backbone stability in *I. tridecemlineatus*.

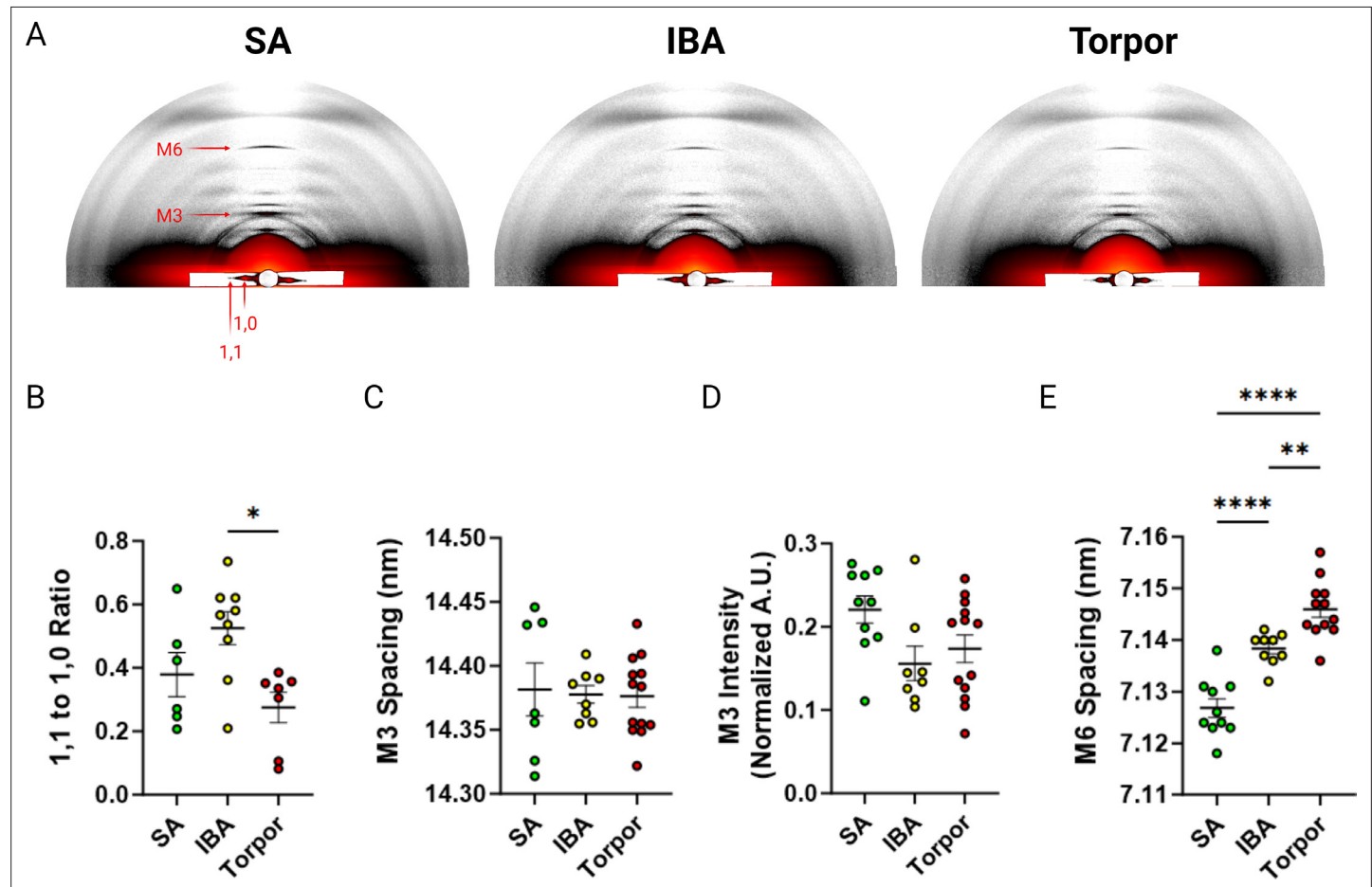

**Figure 3.** X-ray diffraction experiments of skeletal muscle from *Ictidomys tridecemlineaus* demonstrate changes in M6 myosin meridional spacing during torpor. (**A**) Representative X-ray diffraction recordings from permeabilized skeletal muscle bundles from *Ictidomys tridecemlineatus* from summer active (SA), interbout arousal (IBA) and torpor. The M3 and M6 meridional reflections and the 1,0 and 1,1 equatorial reflections are indicated. (**B**) Ratio of the 1,1–1,0 equatorial reflections from active, IBA and torpor skeletal muscle. (**C**) M3 meridional spacing, measured in nm. (**D**) Normalized intensity (A.U.) of the M3 meridional reflection. (**E**) M6 meridional spacing, measured in nm. Colored circles represent the mean value obtained from each skeletal muscle bundle which was recorded. Data is displayed as mean ± SEM. One-way ANOVA was used to calculate statistical significance. *=p < 0.05, **=p < 0.01, ***=p < 0.001. n=5 individual animals per group.

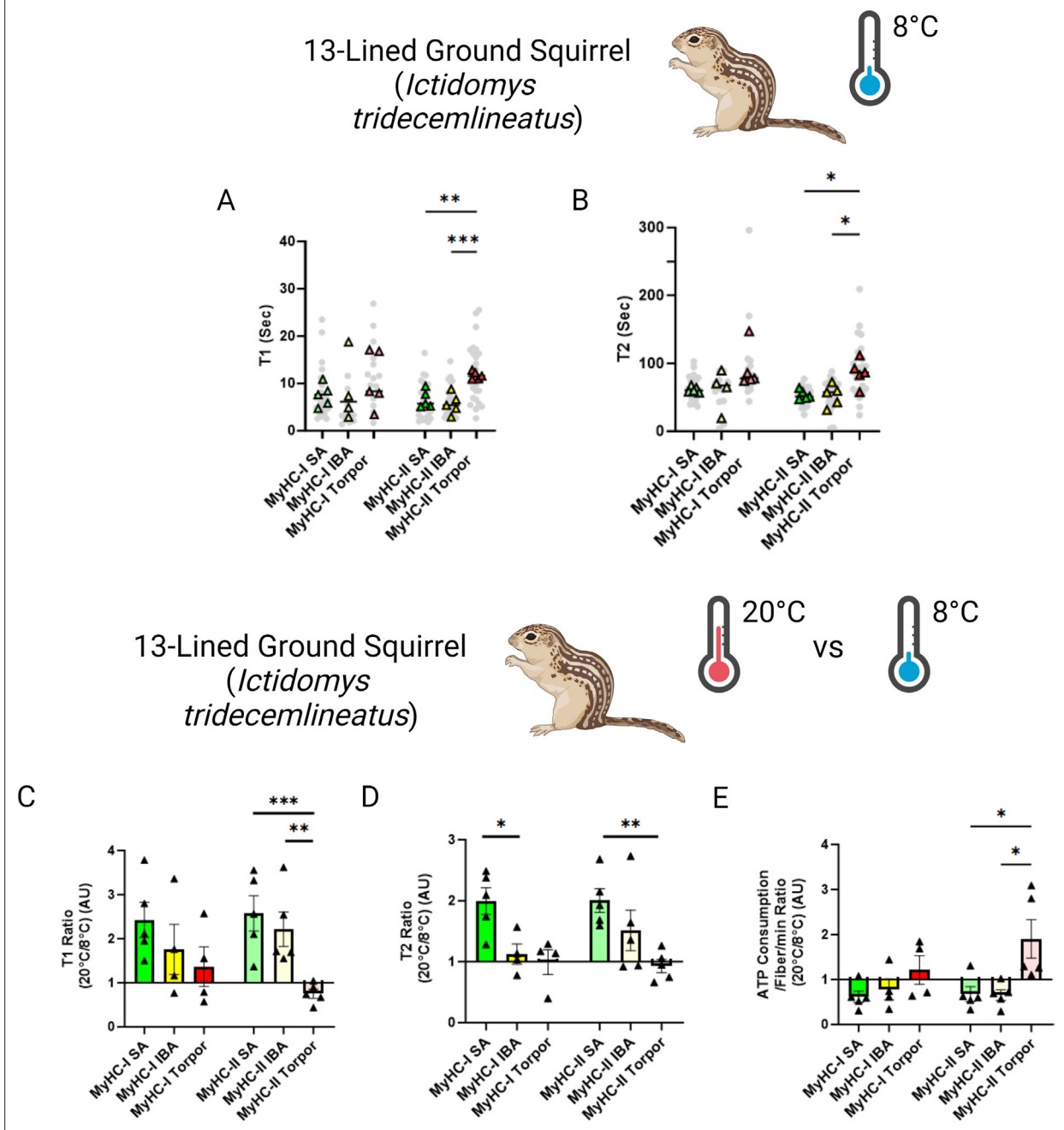

**Figure 4.** Myosin dynamics of *Ictidomys tridecemlineatus* are protected from temperature induced change during torpor, preventing an increase in myosin ATP consumption. (**A**) T1 value in seconds denoting the ATP turnover lifetime of the DRX in *I. tridecemlineatus* at 8 °C. (**B**) T2 value in seconds denoting the ATP turnover lifetime in seconds of the SRX in *I. tridecemlineatus* at 8 °C. (**C**) Ratio of the T1 expressed as the mean value for each matched animal at 20 °C/8 °C, separated for fiber type. (**D**) Ratio of the T2 expressed as the mean value for each matched animal at 20 °C/8 °C, separated for fiber type. (**E**) Ratio of calculated myosin ATP consumption expressed as 20 °C/8 °C, separated for fiber type. Black triangles represent the mean ratio value for each animal. One-way ANOVA was used to calculate statistical significance. *=p < 0.05, **=p < 0.01, ***=p < 0.001. n=5 individual animals per group. Figure created using BioRender.com and published using a CC BY-NC-ND license with permission.

The online version of this article includes the following figure supplement(s) for figure 4:

**Figure supplement 1.** Myosin ATP turnover lifetime is altered following exposure to cold temperature in MyHC-II muscle fibers from *I. tridecemlineatus* during active and IBA periods but not torpor.

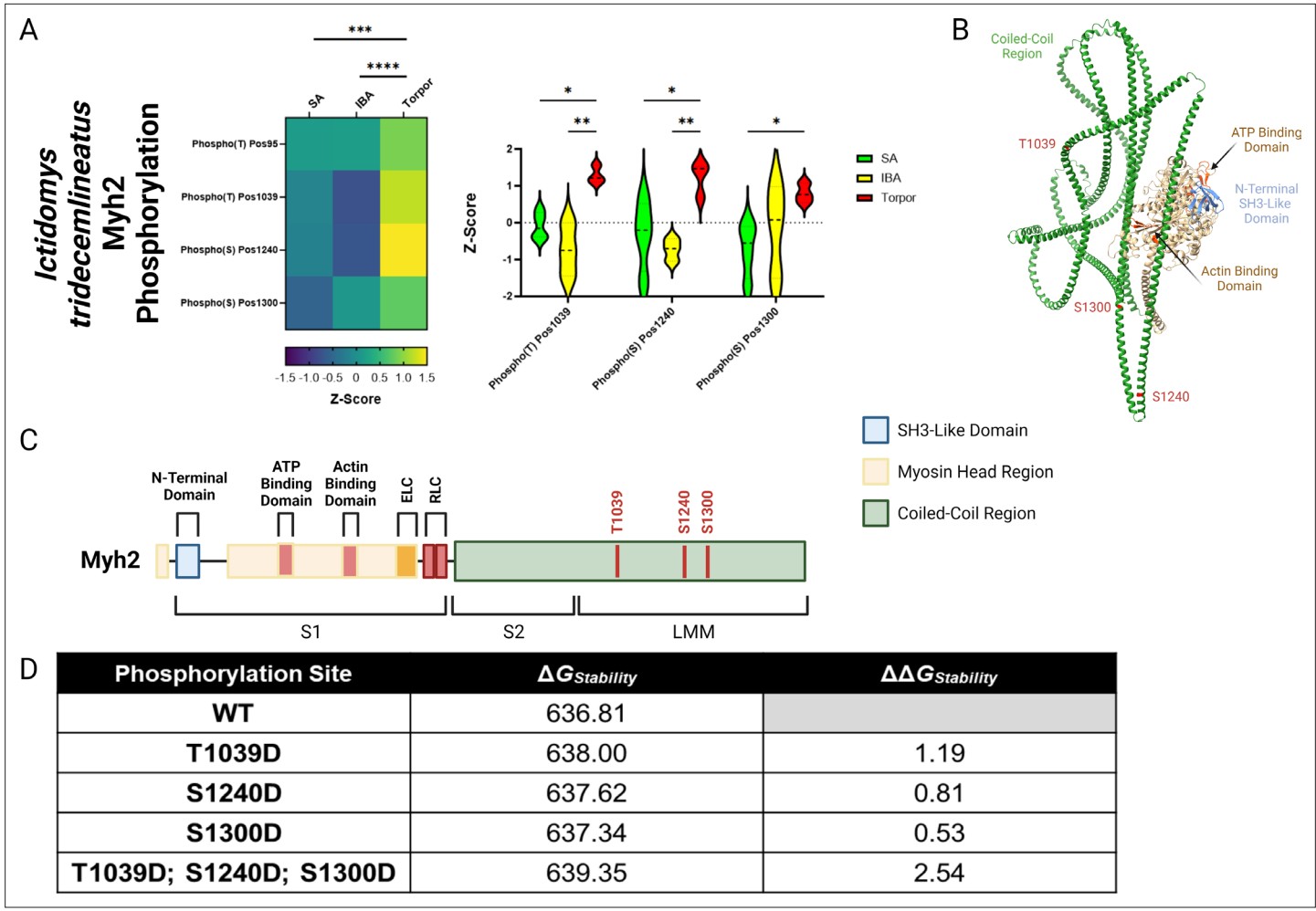

**Figure 5.** MYH2 protein in *Ictidomys tridecemlineatus* is hyper-phosphorylated during torpor, which is predicted to increase protein stability. (**A**) Peptide mapping of differentiated phosphorylation sites upon MYH2 protein during SA, IBA and torpor periods. Heat map demonstrates all sites observed to be differentiated following the calculation of z-scores for each site. Z-scores >0 equal hyper-phosphorylation and z-scores<0 equal hypo-phosphorylation for each residue. Violin plot demonstrates significantly differentiated residues using z-scores. Two-way ANOVA with Šídák's multiple comparisons test was used to calculate statistical significance. *=p < 0.05, **=p < 0.01, ***=p < 0.001, ****=p < 0.0001. n=5 individual animals per group. (**B**) Chimera of MYH2 protein created using ChimeraX software. Important regions of the protein are annotated including coiled-coil region, ATP binding domain, actin binding domain and N-terminal SH3-like domain. Also, significantly hyper-phosphorylated residues are highlighted in red. (**C**) Schematic of MYH2 protein with regions and hyper-phosphorylated resides annotated in red. Figure made in BioRender. (**D**) EvoEF calculations of protein stability in both wild type and phosphor-mimetic mutants. Aspartic acid was used to mimic phospho-threonine/phospho-serine due to their chemical similarity. $\Delta G_{Stability}$ indicates the stability score for the protein in its corresponding configuration. $\Delta\Delta G_{Stability}$ represents the change in stability in mutant proteins versus the wild type protein. $\Delta\Delta G_{Stability}$ of >0 represents an increase in the stability of a mutant versus wild type. Panel C created using BioRender.com and published using a CC BY-NC-ND license with permission.

The online version of this article includes the following figure supplement(s) for figure 5:

**Figure supplement 1.** MYH7 protein phosphorylation and acetylation in *U. arctos* is relatively unchanged during winter periods.

**Figure supplement 2.** MYH2 protein phosphorylation and acetylation in *U*.

## Sarcomeric proteins are dysregulated in resting skeletal myofibers of hibernating *Ictidomys tridecemlineatus*

In addition to PTMs, myosin binding partners and/or surrounding proteins may change during torpor and may contribute to disruptions of myosin metabolic states in *I. tridecemlineatus*. We performed an untargeted global proteomics analysis on isolated muscle fibers. Principal component analyses

showed that whilst SA muscle fibers form a separate entity, both IBA and torpor myofibers are clustered, suggesting a common proteomics signature during the two states of hibernation (*Figure 6A*). Differentially expressed proteins included molecules involved in sarcomere organization and function (e.g. SRX-determining MYBPC2 and sarcomeric scaffold-defining ACTN3) as well as molecules belonging to metabolic pathways (e.g. lipid metabolism-related HMGCS2, carbohydrate metabolism-linked PDK4; *Figure 6B and C* as well as *Figure 6—figure supplements 1 and 2*). Gene ontology analyses including the top five up-/downregulated proteome clusters reinforced the initial findings. They emphasized proteins related to 'muscle system process', 'fiber organization', 'muscular contraction' or 'muscle development' and to 'lipid or carbohydrate metabolism' pathways (*Figure 6D–G*). To complement these results, we performed profiling of core metabolite and lipid contents within the myofibers of *I. tridecemlineatus*. Lipid measurements were significantly lower during IBA, and torpor compared to SA (e.g. omega-6 and omega-7 fatty acids – *Figure 6—figure supplement 3*), in line with previous studies (*Otis et al., 2011*).

Untargeted global proteomics analysis was performed in *U. arctos* where myosin metabolic states are preserved during hibernation (*Figure 1*). As for *I. tridecemlineatus*, the principal component analysis emphasized differences between active and hibernating *U. arctos* (*Figure 6—figure supplement 4*). Differentially expressed proteins, complemented by gene ontology analyses, underscored hibernation-related shifts in metabolic proteins as in *I. tridecemlineatus* (*Figure 6—figure supplement 5*). However, sarcomeric proteins did not appear as differentially expressed molecules, highlighting their potential contribution to the adaptation of myosin in hibernating *I. tridecemlineatus*.

## Discussion

In the present study, our objective was to investigate whether modulating muscle myosin DRX and SRX states could serve as a key mechanism for reducing ATP/metabolic demand during mammalian hibernation. Contrary to our hypothesis, our results indicate that during hibernation small mammals such as *I. tridecemlineatus* or *E. quercinus* modulate their myosin metabolic states unexpectedly by increasing energy expenditure of sarcomeres at ambient temperatures. At 8 °C, muscle fibers from *I. tridecemlineatus* obtained during SA and IBA phases displayed a significant rise in myosin-based ATP consumption. Conversely, fibers sampled during torpor bouts did not exhibit this cold-induced increase. These data suggest that small hibernators may stabilize myosin during torpor to prevent cold-induced increases in energy expenditure and thus increased heat production. Overall, our results also demonstrate a preferential adaptation of type II, fast-twitch, muscle fibers. Type II muscle fibers generally have more plasticity than type I, slow-twitch, fibers and have been demonstrated to undergo behavioral and fiber type transition in response to both metabolic and exercise stimuli (*Qaisar et al., 2016*; *Bourdeau Julien et al., 2018*; *Plotkin et al., 2021*; *Andersen and Aagaard, 2010*).

### Resting myosin conformation is unchanged during hibernation

In contrast to our study hypothesis, we demonstrated that the animals studied continue to maintain their active levels of myosin in the more metabolically active disordered-relaxed state (DRX) during periods of metabolic shutdown. This is of particular interest when compared to a previous study by Toepfer et al., who observed that in cardiac muscle from *I. tridecemlineatus* the percentage of myosin heads in the DRX conformation was lower in periods of torpor vs SA and IBA (*Toepfer et al., 2020*). Maintenance of the resting myosin conformation to active levels during hibernating periods may be to prevent the onset significant muscular atrophy during hibernation. Hibernating animals have evolved mechanisms that prevent skeletal muscle atrophy during the extended periods of immobilization inherent to hibernation (*Hindle et al., 2011*; *Miyazaki et al., 2022*; *Miyazaki et al., 2019*; *Mugahid et al., 2019*). Further research into changes to myosin head conformation in human atrophy and immobilization models would provide an interesting comparison to these data and potentially highlight resting myosin conformation as a novel target in the treatment of sarcopenia associated with aging and/or inactivity. It would furthermore ideally be possible to increase the biological sample size of all the species analysed in this study to further confirm the results which we report, particularly as modest differences are seen in the resting myosin conformation values of *U. americanus*.

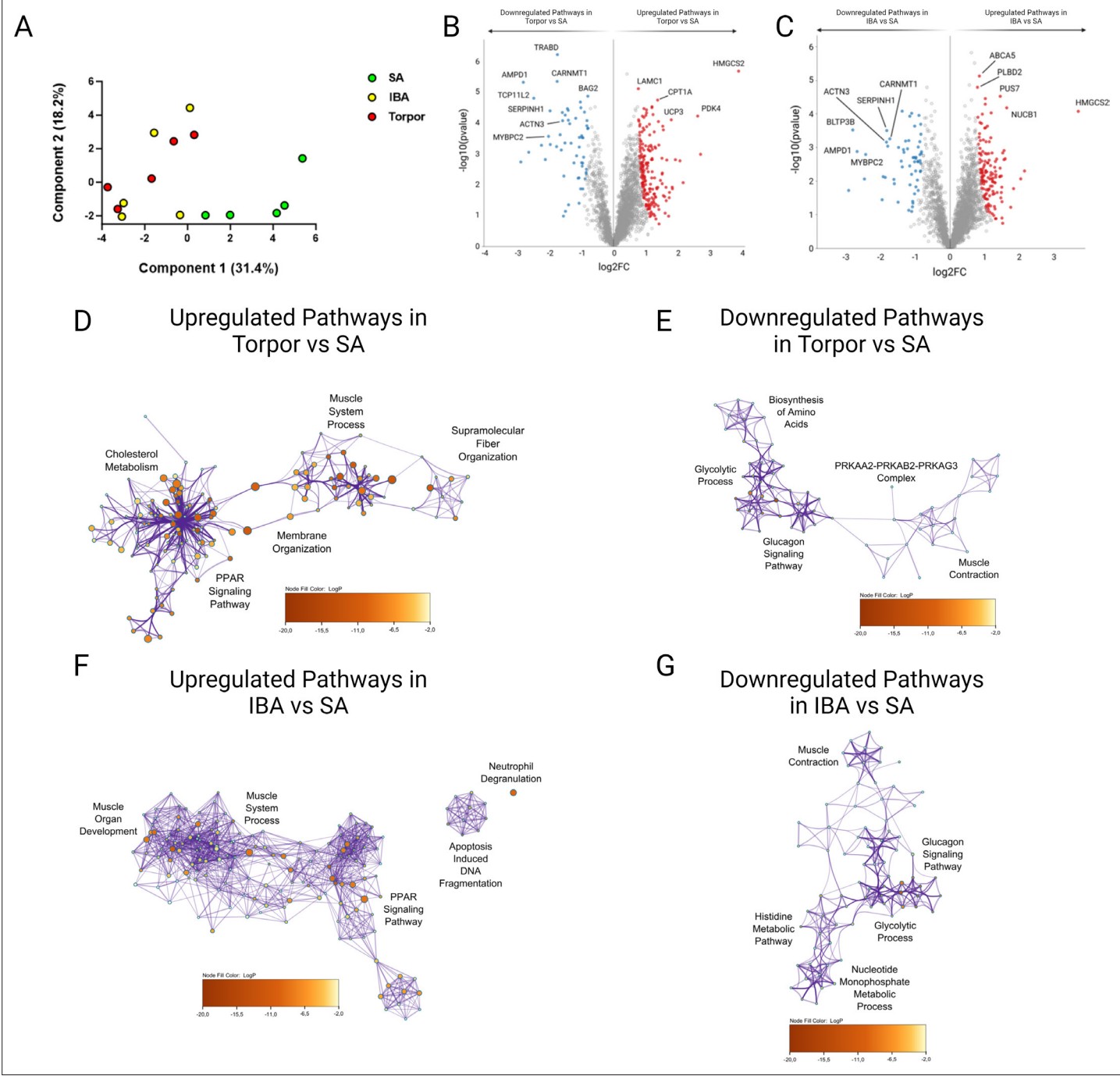

**Figure 6.** Global proteome analysis demonstrates changes to metabolic and sarcomeric changes in skeletal muscle fibers from *Ictidomys tridecemlineatus* during IBA and torpor. (**A**) Principal component analysis for all animals analyzed during SA, IBA and torpor periods. (**B**) Volcano plot displaying proteins which are differentially expressed during torpor vs active periods. FDR < 0.01. Red circles are upregulated proteins and blue circles are downregulated proteins. Highly differentiated proteins of interest are annotated with their respective protein name. (**C**) Volcano plot displaying proteins which are differentially expressed during IBA vs SA periods. Red circles are upregulated proteins and blue circles are downregulated proteins. Highly differentiated proteins of interest are annotated with their respective protein name. (**D**) Ontological associations between proteins upregulated during torpor vs SA periods. The top five association clusters are annotated on the network. A full list of clusters and the proteins lists included in clusters are available in *Figure 6—figure supplement 1* and *Supplementary file 2*. (**E**) Ontological associations between proteins downregulated during torpor vs SA periods. The top five association clusters are annotated on the network. A full list of clusters and the proteins lists included in clusters are available in *Figure 6—figure supplement 1* and *Supplementary file 2*. (**F**) Ontological associations between proteins upregulated during IBA vs SA periods. The top five association clusters are annotated on the network. A full list of clusters and the proteins lists included in clusters are available in *Figure 6—figure supplement 2* and *Supplementary file 3*. (**G**) Ontological associations between proteins downregulated during IBA vs

*Figure 6 continued*

SA periods. The top five association clusters are annotated on the network. A full list of clusters and the proteins lists included in clusters are available in *Figure 6—figure supplement 2* and *Supplementary file 3*. Gene ontology networks were established using Metascape and visualized using Cytoscape. Detailed information upon the statistical testing used is available in the methods section. FDR < 0.01 significantly differentially expressed proteins were used to establish networks. Purple lines indicate a direct interaction. Circle size is determined by enrichment and color is determined by p value. n=5 individual animals per group.

The online version of this article includes the following figure supplement(s) for figure 6:

**Figure supplement 1.** All ontological clusters altered in *I. tridecemlineatus* in torpor vs SA periods.

**Figure supplement 2.** All ontological clusters altered in *I. tridecemlineatus* in IBA vs SA periods.

**Figure supplement 3.** Metabolite and lipid quantification of skeletal muscle from *I.tridecemlineatus* reveals a decrease in lipid levels during torpor.

**Figure supplement 4.** Global proteome analysis of *U.arctos* skeletal muscle fibers reveal metabolic changes but not sarcomeric changes.

**Figure supplement 5.** All ontological clusters altered in *U.arctos* in summer vs winter periods.

## Resting myosin ATP consumption is higher during hibernation in small mammals at ambient temperature

Our findings demonstrate that in small hibernators such as *I. tridecemlineatus* and *E. quercinus*, the ATP turnover time of relaxed myosin molecules (in both DRX and SRX conformations) is faster during torpor (and IBA), especially in type II muscle fibers, leading to an unexpected overall increased ATP consumption. Accordingly, a few studies investigating human pathological conditions have reported disruptions of the myosin ATP turnover times in resting isolated skeletal myofibers, but their actual impacts have never been thoroughly investigated (*Lewis et al., 2023*; *Phung et al., 2020*; *Sonne et al., 2023*). Here, originally, we estimated the consequences on the actual energy consumption of sarcomeres/muscle fibers. Our results of higher ATP consumption during torpor (and IBA) could, at first glance, be seen as counter-intuitive but is an indication of adaptation to the myosin protein during hibernating periods. It was therefore essential to investigate if these changes were also observed at a lower temperature, more relevant to the actual temperature of skeletal muscle in small hibernators during hibernation (*Cooper et al., 2012*).

## Resting myosin ATP turnover time is protected from cold-induced change during torpor in *Ictidomys Tridecemlineatus*

A critical difference between the large hibernators, *U. arctos* and *U. americanus*, and the small hibernators, *E. quercinus* and *I. tridecemlineatus*, during their hibernation periods is core body temperature. Whilst the large hibernators only undergo a modest temperature decrease during hibernation, small hibernators reduce their core body temperature drastically to between 4°C and 8°C (*Cooper et al., 2012*; *Sahdo et al., 2013*; *Kisser and Goodwin, 2012*). We repeated the Mant-ATP chase assays at 8 °C to mimic the environment of physiological torpor. Interestingly, lowering the temperature decreased DRX and SRX-linked ATP turnover times during active periods (in SA and IBA), especially in type II myofibers from *I. tridecemlineatus*, inducing an increase in ATP consumption. Metabolic organs such as skeletal muscle are well-known to increase core body temperature in response to significant cold exposure (i) by inducing rapid involuntary contractions known as shivering (*Haman, 2006*; *Haman and Blondin, 2017*) or (ii) a process named non-shivering thermogenesis (NST). NST has traditionally been attributed to processes in brown adipose tissue, however, in recent years, skeletal muscle has also been shown to contribute to heat production via NST (*Himms-Hagen, 1984*). NST in skeletal muscle is stimulated by $Ca^{2+}$-slippage by the sarcoplasmic reticulum $Ca^{2+}$-ATPase (SERCA) in a cascade of molecular events controlled by a protein called sarcolipin (*Raimbault et al., 2001*; *Nowack et al., 2017*; *Bal and Periasamy, 2020*; *Maurya et al., 2015*). Mammals have evolved a mechanism of resistance to ryanodine receptor (RyR) opening via rises in $Ca^{2+}$ to allow for $Ca^{2+}$ leak which is activated by cAMP via the β-adrenergic system (*Meizoso-Huesca et al., 2022*; *Singh et al., 2023*). This $Ca^{2+}$ slippage leads to the uncoupling of SERCA activity from $Ca^{2+}$ transport across the sarcoplasmic reticulum. Consequently, ATP hydrolysis does not fuel ion transport. Instead, the resultant ADP stimulates heat production via the mitochondrial electron transport system (*Maurya et al., 2015*; *Asahi et al., 2003*). Here we propose that, in addition to SERCA, myosin also contributes to NST in small

hibernators. Interestingly, our group has previously demonstrated that the resting myosin dynamics are altered in patients with RYR1 mutation-related myopathies (**Sonne et al., 2023**).

Of potential further interest to the regulation of myosin would be the differential expression of heat shock proteins (HSPs) during hibernation. Various HSPs have been observed to be differentially expressed during hibernation in mammals such as bears and bats (**Thienel et al., 2023**; **Lee et al., 2008**). This is of relevance to the data from our study as HSPs have been demonstrated to be able to bind sarcomeric proteins and regulate their turnover, including that of myosin itself (**Glazier et al., 2018**; **Ojima et al., 2018**). The proteins they have been shown to interact with include the cardiac isoform of myosin binding protein-C an important regulator of resting myosin conformation in the heart (**McNamara et al., 2017**).

As the biopsies which were used in this study were all obtained from the hind leg of the animals studied, it is important to consider that the myosin dynamics may differ if these biopsies were sampled from different areas in the body. This is particularly important different areas of the body can have different core temperature and in some distal muscles, shivering does not occur (**Aydin et al., 2008**). A study from **Aydin et al., 2008**, demonstrated in mice that when a shivering muscle, soleus, was prevented from undergoing non-shivering thermogenesis via knock-out of UCP1 and were subsequently exposed to cold temperatures, the force production of these muscles was significantly reduced due to prolonged shivering. These results do suggest that even in shivering muscle, non-shivering thermogenesis plays a key role in the generation of heat for survival and for the maintenance of muscle performance. Further work examining potential differences in the resting myosin dynamics of muscles sampled from different sites of the body would be of importance to the field in the future.

Essential to our findings were the simultaneous observations that these cold induced changes in myosin ATP turnover times in each resting myosin state were not observed in *I. tridecemlineatus* samples obtained during torpor. *I. tridecemlineatus* and other similar small hibernators require a significant reduction in their body temperature to survive during winter periods (**Cooper et al., 2012**; **Christian et al., 2014**). Therefore, the inhibition of excess heat production via myosin ATP hydrolysis is likely a protective mechanism which has evolved to facilitate reductions in core body temperature and wider metabolic shutdown during torpor.

## Myh2 is hyper-phosphorylated during torpor increasing protein stability in *Ictidomys tridecemlineatus*

The exact causes of all the above alterations remain unclear but may be linked to unusual PTMs directly targeting MyHCs. Thus, hyper-phosphorylation stabilizing the myosin filament backbone of the Myh2 protein through Thr1039-P, Ser1240-P, and Ser1300-P during torpor is proposed as a main potential underlying biophysical mechanism. Interestingly, in silico molecular dynamics simulations mimicking close-by phosphorylations (Thr1309-P and Ser1362-P) have previously demonstrated a structural impairment of the myosin filament backbone (**Sonne et al., 2023**). This is consistent with our X-ray diffraction experiments where M6 spacing was found to be greater during torpor and is indicative of a unique structural configuration of the thick filament during hibernation (**Ma and Irving, 2022**). Besides PTMs, another potential cause of the myosin metabolic remodeling may be changes to myofibrillar protein expression. Our global untargeted proteomics analysis reveals that *I. tridecemlineatus* undergo a subtle reorganization in sarcomeric protein content (**Hindle et al., 2011**; **Chazarin et al., 2019**; **Miyazaki et al., 2022**). Our results are consistent with previous similar analysis on *I. tridecemlineatus* from Hindle *et. al.,* who identified significant changes in carbohydrate metabolism but also changes in sarcomere and cytoskeletal organization in SA vs torpor muscle (**Hindle et al., 2011**). In our analysis, we observed examples of type II muscle fiber specific proteins which were highly differentially expressed. ACTN3 (α-actinin-3), a type II fiber-specific molecule involved in linking adjacent sarcomeres, was notably downregulated during hibernation (**Wyckelsma et al., 2021**). Its depression in mammals unexpectedly appears to confer superior cold resistance and heat generation in skeletal muscle (**Wyckelsma et al., 2021**; **Pickering and Kiely, 2017**; **Clarkson et al., 2005**). Another type II fiber-specific protein significantly downregulated in both torpor and IBA was MYBPC2 (fast skeletal myosin binding protein-C) (**McNamara et al., 2017**; **Song et al., 2021**). Its loss in skeletal mammals of rodents is thought to directly interfere with myosin conformation (**Song et al., 2021**). Taken together, we believe that the aberrant PTMs and/or protein expression remodeling do play a role in modifying the myosin filament stability and this all may be

triggered by the well-known decreased tension on sarcomeres during hibernation (*Linari et al., 2015*; *Ma et al., 2018b*).

## Conclusion

Our findings suggest ATP turnover adaptations in DRX and SRX myosin states occur in small hibernators like *I. tridecemlineatus* during hibernation and cold exposure. In contrast, larger mammals like *U. arctos* and *U. americanus* show no such changes, likely due to their stable body temperature during hibernation. This supports our hypothesis that myosin serves as a non-shivering thermogenesis regulator in mammals, a mechanism inhibited during torpor.

## Methods

### Samples collection and cryo-preservation

Gastrocnemius muscles from *I. tridecemlineatus* and *E. quercinus* were collected from animals during the summer (when subjects cannot hibernate) as well as from those approximately half-way through a torpor bout and during IBA. Animal husbandry and hibernation status monitoring (using temperature telemetry or biologging) are described elsewhere (*Huber et al., 2021*; *Hutchinson et al., 2022*; *Charlanne et al., 2022*). Tissues were excised and frozen immediately in liquid $N_2$, and subsequently stored at –80 °C. Muscles were shipped from Canada (*I. tridecemlineatus*) or Austria (*E. quercinus*) to Denmark on dry ice. For *E. quercinus*, all procedures have been discussed and approved by the institutional ethics and animal welfare committee in accordance with GSP guidelines and national legislation (ETK-046/03/2020, ETK-108/06/2022), and the national authority according to §§29 of Animal Experiments Act, Tierversuchsgesetz 2012 - TVG 2012 (BMBWF-68.205/0175 V/3b/2018). For *I. tridecemlineatus*, all procedures were approved by the Animal Care Committee at the University of Western Ontario and conformed to the guidelines from the Canadian Council on Animal Care.

In Dalarna, Sweden, subadult (2.5 years to 5.5 years) Scandinavian brown bears (*U.* arctos) were sedated during hibernation and active periods as part of the Scandinavian Brown Bear Project. Each bear was outfitted with GPS collars and VHF transmitters, enabling location tracking in dens during winter and in natural habitats during active months. Bears were located in their dens in late February and again, from a helicopter, in late June. For winter sedation, a cocktail of medetomidine, zolazepam, tiletamine, and ketamine was used. During summer captures, bears were sedated from a helicopter using a higher dose of medetomidine, zolazepam, and tiletamine, omitting ketamine, to adjust for increased metabolic activity (*Evans et al., 2012*). All experiments on brown bears were performed with approval by the Swedish Ethical Committee on Animal Research (C18/15 and C3/16).

Protocols for black bear experiments were approved by the University of Alaska Fairbanks, Institutional Animal Care and Use Committee (IACUC nos. 02–39, 02–44, 05–55, and 05–56). Animal work was carried out in compliance with the IACUC protocols and ARRIVE guidelines. Animal care, monitoring of physiological conditions of the black bear (*U. americanus*) and tissue harvesting were described previously (*Fedorov et al., 2014*). Before tissue sampling, bears (51–143 kg) were captured in the field by Alaska Department of Fish and Game in May–July and kept in an outdoor enclosure for at least 2 months to allow adaptation for changes in mobility. Feeding was stopped 24 hr before summer active animals were euthanized. Hibernating bears were without food or water since October 27 and euthanized for tissue sampling between March 1 and 26, about 1 month before expected emergence from hibernation. Core body temperature was recorded with radio telemetry and oxygen consumption and respiratory quotient were monitored in hibernating bears with open flow respirometry. Samples of quadriceps muscle have been collected from captive hibernating and summer active males older than 2 years and banked at –80 °C.

### Solutions

As previously published [79, 80], the relaxing solution contained 4 mM Mg-ATP, 1 mM free $Mg^{2+}$, $10^{-6}$ mM free $Ca^{2+}$, 20 mM imidazole, 7 mM EGTA, 14.5 mM creatine phosphate and KCl to adjust the ionic strength to 180 mM and pH to 7.0. Additionally, the rigor buffer for Mant-ATP chase experiments contained 120 mM K acetate, 5 mM Mg acetate, 2.5 mM $K_2HPO_4$, 50 mM MOPS, 2 mM DTT with a pH of 6.8.

## Muscle preparation and fibre permeabilization

Cryopreserved muscle samples were dissected into small sections and immersed in a membrane-permeabilising solution (relaxing solution containing glycerol; 50:50 v/v) for 24 hr at −20°C, after which they were transferred to 4°C. These bundles were kept in the membrane-permeabilising solution at 4°C for an additional 24 hr. After these steps, bundles were stored in the same buffer at −20°C for use up to 1 week (*Ross et al., 2019*; *Ross et al., 2020*).

## Mant-ATP chase experiments

On the day of the experiments, bundles were transferred to the relaxing solution and individual muscle fibres were isolated. Their ends were individually clamped to half-split copper meshes designed for electron microscopy (SPI G100 2010C-XA, width, 3 mm), which had been glued to glass slides (Academy, 26 x 76 mm, thickness 1.00–1.20 mm). Cover slips were then attached to the top to create a flow chamber (Menzel-Gläser, 22 x 22 mm, thickness 0.13-0.16 mm) (*Ochala et al., 2021*; *Ranu et al., 2022*). Subsequently, at 20°C, myofibers with a sarcomere length of 2.00 µm were kept (assessed using the brightfield mode of a Zeiss Axio Scope A1 microscope). Each muscle fibre was first incubated for 5 min with a rigor buffer. A solution containing the rigor buffer with added 250 µM Mant-ATP was then flushed and kept in the chamber for 5 min. At the end of this step, another solution made of the rigor buffer with 4 mM ATP was added with simultaneous acquisition of the Mant-ATP chase.

For fluorescence acquisition, a Zeiss Axio Scope A1 microscope was used with a Plan-Apochromat 20x/0.8 objective and a Zeiss AxioCam ICm 1 camera. Frames were acquired every five seconds with a 20 ms acquisition/exposure time using at 385nm, for 5 min. Three regions of each individual myofiber were sampled for fluorescence decay using the ROI manager in ImageJ as previously published (*Ochala et al., 2021*; *Ranu et al., 2022*). The mean background fluorescence intensity was subtracted from the average of the fibre fluorescence intensity for each image. Each time point was then normalized by the fluorescence intensity of the final Mant-ATP image before washout (T = 0). These data were then fit to an unconstrained double exponential decay using Graphpad Prism 9.0:

$$\text{Normalized Fluorescence} = 1 - P1(1 - \exp^{(-t/T1)}) - P2(1 - \exp^{(-t/T2)})$$

where P1 (DRX) is the amplitude of the initial rapid decay approximating the disordered-relaxed state with T1 as the time constant for this decay. P2 (SRX) is the slower second decay approximating the proportion of myosin heads in the super-relaxed state with its associated time constant T2 (*Ochala et al., 2021*).

Mant-ATP Chase Experiment were performed at ambient lab temperature (20 °C) for all samples unless otherwise stated. An additional setup was made to do the experiments at colder temperatures (~8 °C). All slides were prepared in the same manner as mentioned above until incubation with rigor buffer and Mant-ATP buffer. All slides were kept on a metal plate on ice while incubated with both rigor and Mant-ATP buffer. The temperature of the chambers was measured (~8 ) before flushing with ice-cold ATP.

## Fiber-type staining

Fiber typing After the completion of the Mant-ATP chase experiments, individual fibers were stained with an anti-MyHC slow/type I antibody (A4.951; IgM isoform: 1:50, DSHB). Fibers were then washed in PBS and incubated with a secondary antibody conjugated to Alexa 647 in a goat serum (Thermo Fisher Scientific, dilution 1:1000). After washing, the muscle fibers were mounted in Fluoromount, and images were taken with a Zeiss AXIO Lab A1 microscope (Carl Zeiss AG, GE, objectives × 20 and×10). Positive staining with the MyHC β-slow/type I antibody indicated a type I muscle fiber and negative staining with the MyHC β-slow/type I antibody indicated a type II muscle fiber. Comparisons between muscle fibers sampled in summer or winter were then separated accordingly. Fiber-type breakdown and analysis for all samples used in this study are shown in *Supplementary file 1*.

## X-ray diffraction recordings and analyses

Thin muscle bundles were mounted and transferred to a specimen chamber which was filled with the relaxing buffer. The ends of these thin muscle bundles were then clamped at a sarcomere length of 2.00 µm. Subsequently, X-ray diffraction patterns were recorded at 15°C using a CMOS camera (Model C11440-22CU, Hamamatsu Photonics, Japan, 2048 x 2048 pixels) in combination with a 4-inch

image intensifier (Model V7739PMOD, Hamamatsu Photonics, Japan). The X-ray wavelength was 0.10 nm and the specimen-to-detector distance was 2.14 m. For each preparation, approximately 20–50 diffraction patterns were recorded at the BL40XU beamline of SPring-8 and were analyzed as described previously (*Ochala et al., 2010*). To minimize radiation damage, the exposure time was kept low (0.5 or 1 s) and the specimen chamber was moved by 100 µm after each exposure. Following X-ray recordings, background scattering was subtracted, and the major myosin meridional reflection intensities/spacing were determined as described elsewhere previously (*Hessel et al., 2022*; *Ochala et al., 2023*).

## Myosin heavy chain band Isolation for post-translational identifications

Briefly, muscle biopsy samples were cut into 15 mg sections. These were immersed into a sample buffer (0.5 M Tris pH = 6.8, 0.5 mg/ml Bromophenol Blue, 10% SDS, 10% Glycerol, 1.25% Mercaptoethanol) at 4 °C. Samples were then homogenized and centrifuged allowing the supernatant to be extracted and used for SDS-PAGE gels (with stacking gel made with Acrylamide/Bis 37.5:1 and separation gel made with Acrylamide/Bis 100:1). Proteins were separated and individual MyHC bands excised (*Andersen and Aagaard, 2000*).

## Post-translational modification peptide mapping

Proteins were separated on an SDS-PAGE gel (6% polyacrylamide and 30% glycerol), and individual bands excised. For both *U. arctos* and *I. tridecemlineatus*, separate gel bands were excised for Myh7 and Myh2 and their relevant molecular weights. Gel bands were destained twice with destaining buffer (25 mM ammonium bicarbonate, 50% acetonitrile), dehydrated with 100% acetonitrile, and incubated with reduction/alkylation solution (50 mM Tris pH = 8.5, 10 mM Tcep, 40 mM CAA) for 15 min at 37 °C. Gel bands were washed in destaining buffer, dehydrated, and incubated with 500 ng Trypsin in 20 µL digestion buffer (50 mM TEAB) for 15 min at 37 °C. A total of 30 µL additional digestion buffer was added, and the gel bands incubated over night at 37 °C. After collection of the digested peptides, gel bands were eluted once in 50 µl 1% TFA. Both eluates were combined, and peptides desalted on C18 material prior to LC-MS analysis.

Liquid chromatography was performed using a Vanquish Neo HPLC system (Thermo Fisher Scientific) coupled through a nano-electrospray source to a Tribrid Ascend mass spectrometer (Thermo Fisher Scientific). Peptides were loaded in buffer A (0.1% formic acid) and separated on a 25 cm column Aurora Gen2, 1.7uM C18 stationary phase (IonOpticks) with a non-linear gradient of 1–48% buffer B (0.1% formic acid, 99.9% acetonitrile) at a flow rate of 400 nL/min over 53 min. The column temperature was kept at 50 ° C. Spray voltage was set to 2200 V. Data acquisition switched between a full scan (60 K resolution, 123ms max. injection time, AGC target 100 %) and 10 data-dependent MS/MS scans (30 K resolution, 59ms maximum injection time, AGC target 400% and HCD activation type). Isolation window was set to 1.4, and normalized collision energy to 25. Multiple sequencing of peptides was minimized by excluding the selected peptide candidates for 45 s.

## Global proteome profiling

Fresh collagenase dilution in DMEM was first prepared by filtering the collagenase through 22 µm. The solution is placed in a+37 chamber before use. Two g of each snap frozen sample is added to Eppendorf tubes with marked sample ID. In each tube, 200 µl of +37 Collagenase is added to break down the connected tissue in the muscle sample to prevent connective tissue in the sample to be analyzed. All tubes were incubated at +37 for 90 min and agitated every 15 min. After this treatment, samples were each put in a six-well plate. Fibers were then cleaned from connective tissue and separated with tweezers. About 50 fibers of each sample were transferred into an Eppendorf tube with ice-cold PBS. Fibers were spun down at 400 × *g* and 4 and PBS was removed. Skeletal muscle fiber tissue samples were then lysed with lysis buffer (1% (w/v) Sodium Deoxycholate, 100 mM Teab, pH 8.5) and incubated for 10 min at 95 °C followed by sonication using a Bioruptor pico (30 cycles, 30 s on/off, ultra-low frequency). Heat incubation and sonication were repeated once, samples cleared by centrifugation, reduced with 5 mM (final concentration) of TCEP for 15 min at 55 °C, alkylated with 20 mM (final concentration) CAA for 30 min at RT, and digested adding Trypsin/LysC at 1:100 enzyme/ protein ratio. Peptides were cleaned up using StageTips packed with SDB-RPS and resuspended in 50 µL TEAB 100 mM, pH 85. A total of 50 µg of each sample was labeled with 0.5 mg TMTpro labeling

reagent according to the manufacturer's instructions. Labeled peptides were combined and cleaned up using C18-E (55 µm, 70 Å, 100 mg) cartridges (Phenomenex).

Labeled desalted peptides were resuspended in buffer A* (5% acetonitrile, 1% TFA), and fractionated into 16 fractions by high-pH fractionation. For this, 20 µg peptides were loaded onto a Kinetex 2.6u EVO C18 100 Å 150 × 0.3 mm column via an EASY-nLC 1200 HPLC (Thermo Fisher Scientific) in buffer AF (10 mM TEAB), and separated with a non-linear gradient of 5–44% buffer BF (10 mM TEAB, 80% acetonitrile) at a flow rate of 1.5 µL / min over 62 min. Fractions were collected every 60 s with a concatenation scheme to reach 16 final fractions (e.g. fraction 17 was collected together with fraction 1, fraction 18 together with fraction 2, and so on).

Fractions were evaporated, resuspended in buffer A*, and measured on a Vanquish Neo HPLC system (Thermo Fisher Scientific) coupled through a nano-electrospray source to a Tribrid Ascend mass spectrometer (Thermo Fisher Scientific). Peptides were loaded in buffer A (0.1% formic acid) onto a 110 cm mPAC HPLC column (Thermo Fisher Scientific) and separated with a non-linear gradient of 1–50% buffer B (0.1% formic acid, 80% acetonitrile) at a flow rate of 300 nL/min over 100 min. The column temperature was kept at 50 ° C. Samples were acquired using a Real Time Search (RTS) MS3 data acquisition where the Tribrid mass spectrometer was switching between a full scan (120 K resolution, 50ms max. injection time, AGC target 100%) in the Orbitrap analyzer, to a data-dependent MS/MS scans in the Ion Trap analyzer (Turbo scan rate, 23ms maximum injection time, AGC target 100% and HCD activation type). Isolation window was set to 0.5 (m/z), and normalized collision energy to 32. Precursors were filtered by charge state of 2–5 and multiple sequencing of peptides was minimized by excluding the selected peptide candidates for 60 s. MS/MS spectra were searched in real time on the instrument control computer using the Comet search engine with either the UP000291022 *U. americanus* or UP000005215 *I. tridecemlineatus* FASTA file, 0 max miss cleavage, 1 max oxidation on methionine as variable mod. and 35ms max search time with an Xcorr soring threshold of 1.4 and 20 precursor ppm error. MS/MS spectra resulting in a positive RTS identification were further analyzed in MS3 mode using the Orbitrap analyzer (45 K resolution, 105ms max. injection time, AGC target 500%, HCD collision energy 55 and SPS = 10). The total fixed cycle time, switching between all hree MS scan types, was set to 3 s.

## Proteomics data analysis

Raw mass spectrometry data from peptide mapping experiments were analyzed with MaxQuant (v2.1.4). Peak lists were searched against the *U. americanus* (Uniprot UP000291022) or *I. tridecemlineatus* (Uniprot UP000005215) proteomes combined with 262 common contaminants by the integrated Andromeda search engine. False discovery rate was set to 1% for both peptides (minimum length of 7 amino acids) and proteins. Phospho(STY) and Acetyl(K) were selected as variable modifications.

Raw mass spectrometry data from global proteome profiling were analyzed with Proteome Discoverer (v3.0.1.27) using the default processing workflow 'PWF_Tribrid_TMTpro_SPS_MS3_SequestHT_ INFERYS_Rescoring_Percolator'. Briefly, peak lists were searched against the UniProtKB UP000291022 *U. americanus* or UP000005215 *I. tridecemlineatus* FASTA databases by the integrated SequestHT search engine, setting Carbamidomethyl (C) and TMTpro (K, N-Term) as static modifications, Oxidation (M) as variable modification, max missed cleavage as 2 and minimum peptide amino acid length as 7. The false discovery rate was set to 0.01 (strict) and 0.05 (relaxed).

All statistical analysis of TMT derived protein expression data was performed using in-house developed python scripts based on the analysis pipeline of the Clinical Knowledge Graph (*Santos et al., 2022*). Protein abundances were log2-transformed, and proteins with less than two valid values in at least one group were excluded from the analysis. Missing values were imputed with MinProb approach (width = 0.3 and shift = 1.8). Statistically significant proteins were determined by unpaired t-tests, with Benjamini-Hochberg correction for multiple hypothesis testing. Fold-change (FC) and False Discovery Rate (FDR) thresholds were set to 2 and 0.05 (5%), respectively.

The mass spectrometry proteomics data have been deposited to the ProteomeXchange Consortium via the PRIDE partner repository (*Perez-Riverol et al., 2022*) with the dataset identifier PXD044505, PXD044685, and PXD044728.

Gene Ontology networks were created using Metascape software (*Zhou et al., 2019*). Networks were annotated using Cytoscape (*Shannon et al., 2003*). Principal component plots analysis were created in Perseus (*Tyanova et al., 2016*).

## ¹H-NMR metabolomic characterization

Frozen lyophilized tissue samples were shipped on dry ice to Biosfer Teslab (Reus, Spain) for the ¹H-NMR analysis. Prior to analysis, aqueous and lipid extracts were obtained using the Folch method with slight modifications (*Folch et al., 1957*). Briefly, 1440 µL of dichloromethane:methanol (2:1, v/v) were added to 25 mg of pulverized tissue followed by three 5 min sonication steps with one shaking step in between. Next, 400 µL of ultrapure water was added, mixed, and centrifuged at 25,100 × *g* during 5 min at 4 °C. Aqueous and lipid extracts were transferred to a new Eppendorf tube and completely dried in SpeedVac to achieve solvent evaporation and frozen at –80 °C until ¹H-NMR analysis.

Aqueous extracts were reconstituted in a solution of 45 mM PBS containing 2.32 mM of Trimethylsilylpropanoic acid (TSP) as a chemical shift reference and transferred into 5 mm NMR glass tubes. ¹H-NMR spectra were recorded at 300 K operating at a proton frequency of 600.20 MHz using an Avance III-600 Bruker spectrometer. One-dimensional ¹H pulse experiments were carried out using the nuclear Overhauser effect spectroscopy (NOESY)-presaturation sequence to suppress the residual water peak at around 4.7 ppm and a total of 64 k data points were collected. The acquired spectra were phased, baseline-corrected and referenced before performing the automatic metabolite profiling of the spectra datased through and adaptation of Dolphin (*Gómez et al., 2014*). Several database engines (Bioref AMIX database Bruker), Chenomx and HMDB, and literature were used for 1D-resonances assignment and metabolite identification (*Wishart et al., 2022*; *Vinaixa et al., 2010*).

Lipid extracts were reconstituted in a solution of $CDCl_3:CD_3OD:D_2O$ (16:7:1, v/v/v) containing Tetramethylsilane (TMS) and transferred into 5 mm NMR glass tubes. ¹H-NMR spectra were recorded at 286 K operating at a proton frequency of 600.20 MHz using an Avance III-600 Bruker spectrometer. A 90° pulse with water pre-saturation sequence (ZGPR) was used. Quantification of lipid signals in 1H-NMR spectra was carried out with LipSpin an in-house software based on Matlab (*Barrilero et al., 2018*). Resonance assignments were done based on literature values (*Vinaixa et al., 2010*).

## Myosin chimera simulation

ChimeraX was used to make a simulation of Myh2 to illustrate the changes occurring during torpor in *I. tridecemlineatus*. The sequence was downloaded from UniProt, and significant post-translational modifications positions were highlighted on the protein simulation and marked with red. ATP binding domain and actin binding domain were additionally highlighted.

## EvoEF protein stability simulations

EvoEF (version 1) was used to calculate the stability change upon mutation, in terms of ΔΔG. To this end, we first used 'EvoEF --`command=RepairStructure`' to repair clashes and torsional angles of the wild type structure. 'EvoEF --`command=BuildMutant`' is then used to mutate the repaired wild type structures into the mutant by changing the side chain amino acid type followed by a local side chain repacking. 'EvoEF --`command=ComputeStability`' is then applied to both the repair wild type and the mutant to calculate their respective stabilities (ΔGWT and ΔGmutant). The stability change upon mutation can then be derived by ΔΔG=ΔGmutant-ΔGWT. A ΔΔG below zero means that the mutation causes destabilization; otherwise, it induces stabilization (*Huang et al., 2020*; *Pearce et al., 2019*). The sequence of the MYH2 coiled-coil backbone was used for EvoEF stability calculations due to size limitations in the software when using the entire MYH2 protein sequence.

## Statistical analysis

Data are presented as means ± standard deviations. Statistical tests used are listed in the figure legends.Graphs were prepared and analysed in Graphpad Prism v9. Statistical significance was set to $p < 0.05$ unless otherwise stated.

# Acknowledgements

This work was generously funded by the Carlsberg Foundation (CF20-0113) grant to JO The X-ray experiments were performed under approval of the SPring-8 Proposal Review Committee (2022A1069 and 2022B1107). The related X-ray data reduction and analyses were performed by Accelerated Muscle Biotechnologies Consultants LLC (USA). Mass spectrometry analyses were performed by the Proteomics Research Infrastructure (PRI) at the University of Copenhagen, supported by the Novo

Nordisk Foundation (grant agreement number NNF19SA0059305). The Scandinavian brown bear research project is funded by the Norwegian Environment Agency and the Swedish Environmental Protection Agency. KLD, AVG and VBF were supported by P20GM130443. Tissue collection from I. tridecemlineatus was supported by a Discovery Grant to JFS from the Natural Sciences and Engineering Research Council (Canada).

## Additional information

### Competing interests

Anthony L Hessel: ALH is an owner of Accelerated Muscle Biotechnologies Consultants LLC, which performed the X-ray data reduction and analysis, but services rendered were not linked to outcome or interpretation. The other authors declare that no competing interests exist.

### Funding

| Funder | Grant reference number | Author |
| --- | --- | --- |
| Carlsbergfondet | CF20-0113 | Julien Ochala |
| Novo Nordisk Foundation | NNF19SA0059305 | Julien Ochala |
| Norwegian Environment Agency and the Swedish Environmental Protection Agency | P20GM130443 | Kelly Drew Anna V Goropashnaya Vadim B Fedorov |
| Natural Sciences and Engineering Research Council (Canada) | | James F Staples |

The funders had no role in study design, data collection and interpretation, or the decision to submit the work for publication.

### Author contributions

Christopher TA Lewis, Conceptualization, Data curation, Formal analysis, Supervision, Validation, Investigation, Visualization, Methodology, Writing – original draft, Project administration, Writing – review and editing; Elise G Melhedegaard, Data curation, Formal analysis, Methodology, Writing – review and editing; Marija M Ognjanovic, Mathilde S Olsen, Jenni Laitila, Magnus Gronset, Changxin Zhang, Hiroyuki Iwamoto, Data curation, Writing – review and editing; Robert AE Seaborne, Anthony L Hessel, Michel N Kuehn, Formal analysis, Writing – review and editing; Carla Merino, Nuria Amigo, Data curation, Formal analysis, Writing – review and editing; Ole Frobert, Sylvain Giroud, James F Staples, Anna V Goropashnaya, Vadim B Fedorov, Brian Barnes, Oivind Toien, Kelly Drew, Ryan J Sprenger, Resources, Investigation, Writing – review and editing; Julien Ochala, Conceptualization, Resources, Formal analysis, Supervision, Funding acquisition, Visualization, Methodology, Writing – original draft, Project administration, Writing – review and editing

### Author ORCIDs

Christopher TA Lewis ⓘ https://orcid.org/0000-0003-4477-422X
Sylvain Giroud ⓘ http://orcid.org/0000-0001-6621-7462
Oivind Toien ⓘ http://orcid.org/0000-0001-5967-2483

### Ethics

For E. quercinus, all procedures have been discussed and approved by the institutional ethics and animal welfare committee in accordance with GSP guidelines and national legislation (ETK-046/03/2020, ETK-108/06/2022), and the national authority according to §§29 of Animal Experiments Act, Tierversuchsgesetz 2012 - TVG 2012 (BMBWF-68.205/0175-V/3b/2018). For I. tridecemlineatus all procedures were approved by the Animal Care Committee at the University of Western Ontario and conformed to the guidelines from the Canadian Council on Animal Care.All experiments on brown bears were performed with approval by the Swedish Ethical Committee on Animal Research (C18/15 and C3/16).Protocols for black bear experiments were approved by the University of Alaska Fairbanks,

Institutional Animal Care and Use Committee (IACUC nos. 02-39, 02-44, 05-55, and 05-56). Animal work was carried out in compliance with the IACUC protocols and ARRIVE guidelines.

Reviewer #1 (Public Review): https://doi.org/10.7554/eLife.94616.3.sa1
Reviewer #2 (Public Review): https://doi.org/10.7554/eLife.94616.3.sa2
Reviewer #3 (Public Review): https://doi.org/10.7554/eLife.94616.3.sa3
Author response https://doi.org/10.7554/eLife.94616.3.sa4

## Additional files

### Supplementary files

• Supplementary file 1. Fiber type compositions from animals used for this study. Table demonstrating the percentage of fibers which were analyzed during Mant-ATP chase assays that were either MyHC type I or MyHC type II. Data is presented as mean for each animal ± SD. One-way ANOVA was used to calculate significance between hibernation periods in *I. tridecemlineatus* and *E. quercinus*. Student's t-test was used to calculate significant between hibernating periods in *U. arctos* and *U. americanus* and between MyHC type I and MyHC type II in all animals. ##=$p < 0.01$ vs MyHC type I in corresponding group. ####=$p < 0.0001$ vs MyHC type I in corresponding group. n=5 individual animals per group.

• Supplementary file 2. Detailed protein lists for top five differentially expressed ontological clusters in *I. tridecemlineatus* in torpor vs SA periods. **A**. Table details the list of proteins which were found to be differentially upregulated in torpor vs SA in each corresponding ontological cluster as identified by Metascape. Clusters are arranged by order of statistical significance. Proteins are listed in alphabetical order within each cluster. **B**. Table details the list of proteins which were found to be differentially downregulated in torpor vs SA in each corresponding ontological cluster as identified by Metascape. Clusters are arranged by order of statistical significance. Proteins are listed in alphabetical order within each cluster. n=5 individual animals per group.

• Supplementary file 3. Detailed protein lists for top five differentially expressed ontological clusters in *I. tridecemlineatus* in IBA vs SA periods. **A**. Table details the list of proteins which were found to be differentially upregulated in IBA vs SA in each corresponding ontological cluster as identified by Metascape. Clusters are arranged by order of statistical significance. Proteins are listed in alphabetical order within each cluster. **B**. Table details the list of proteins which were found to be differentially downregulated in IBA vs SA in each corresponding ontological cluster as identified by Metascape. Clusters are arranged by order of statistical significance. Proteins are listed in alphabetical order within each cluster. n=5 individual animals per group.

• Supplementary file 4. Detailed protein lists for top five differentially expressed ontological clusters in *U. arctos* in winter vs summer periods. **A**. Table details the list of proteins which were found to be differentially upregulated in winter vs summer in each corresponding ontological cluster as identified by Metascape. Clusters are arranged by order of statistical significance. Proteins are listed in alphabetical order within each cluster. **B**. Table details the list of proteins which were found to be differentially downregulated in winter vs summer in each corresponding ontological cluster as identified by Metascape. Clusters are arranged by order of statistical significance. Proteins are listed in alphabetical order within each cluster. n=5 individual animals per group.

• MDAR checklist

### Data availability

Mass Spectrometry proteomics data have been deposited to the ProteomeXchange Consortium via the PRIDE partner repository with the dataset identifiers PXD044505, PXD044685 and PXD044728.

The following datasets were generated:

| Author(s) | Year | Dataset title | Dataset URL | Database and Identifier |
|---|---|---|---|---|
| Wierer M, Ochala J | 2024 | Proteomic analysis of skeletal muscle fibre samples from hibernating and awake bears | http://www.ebi.ac.uk/pride/archive/projects/PXD044505 | PRIDE, PXD044505 |
| Wierer M, Ochala J | 2024 | Myosin PTM peptide mapping from hibernating and active bears and squirrels | http://www.ebi.ac.uk/pride/archive/projects/PXD044685 | PRIDE, PXD044685 |
| Wierer M, Ochala J | 2024 | Proteomic analysis of skeletal muscle fibre samples from hibernating and awake squirrels | http://www.ebi.ac.uk/pride/archive/projects/PXD044728 | PRIDE, PXD044728 |

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
