## [Editor Report · eLife assessment]

The work by Lewis and co-workers presents **important** findings on the role of myosin structure/energetics on the molecular mechanisms of hibernation by comparing muscle samples from small and large hibernating mammals. The **solid** methodological approaches have revealed insights into the mechanisms of non-shivering thermogenesis and energy expenditure.

---

## [Referee Report · Reviewer #1 (Public Review)]

Summary:

The evolution of non-shivering thermogenesis is of fundamental importance to understand. Here, in small mammals the contractile apparatus of the muscle are shown to increase energy expenditure upon a drop in ambient temperature. Additionally, in the state of torpor, small hibernators did not show an increase in energy expenditure under the same challenge.

Strengths:

The authors have conducted a very well-planned study that has sampled the muscle of large and small hibernators from two continents. Multiple approaches were then used to identify the state of the contractile apparatus, and its energy expenditure under torpor or otherwise.

Weaknesses:

There was only one site of biopsy from the animals used (leg). As the authors state, it would be interesting to know if non-shivering thermogenesis is something that is regionally different in the animal, given the core body and distal limbs have different temperatures.

---

## [Referee Report · Reviewer #2 (Public Review)]

Summary:

The authors utilized (permeabilized) fibers from muscle samples obtained from brown and black bears, squirrels, and Garden dormice, to provide interesting and valuable data regarding changes in myosin conformational states and energetics during hibernation and different types of activity in summer and winter. Assuming that myosin structure is similar between species then its role as a regulator of metabolism would be similar and not different, yet the data reveal some interesting and perplexing differences between the selected hibernating species.

Strengths:

The experiments on the permeabilized fibers are complementary, sophisticated, and well-performed, providing new information regarding the characteristics of skeletal muscle fibers between selected hibernating mammalian species under different conditions (summer, interarousal, and winter).

The studies involve complementary assessments of muscle fiber biochemistry, sarcomeric structure using X-ray diffraction, and proteomic analyses of posttranslational modifications.

Weaknesses:

It would be helpful to put these findings on permeabilized fibers into context with the other anatomical/metabolic differences between the species to determine the relative contribution of myosin energetics (with these other contributors) to overall metabolism in these different species, including factors such as fat volume/distribution.

---

## [Referee Report · Reviewer #3 (Public Review)]

Summary and Strengths:

The manuscript by Lewis et al, investigates whether myosin ATP activity may differ between states of hibernation and activity in both large and small mammals. The study interrogates (primarily) permeabilized muscle strips or myofibrils using several state-of-the-art assays, including the mant-ATP assay to investigate ATP utilization of myosin, X-ray diffraction of muscles, proteomics studies, metabolic tests, and computational simulations. The overall data suggests that ATP utilization of myosin during hibernation is different than in active conditions.

A clear strength of this study is the use of multiple animals that utilize two different states of hibernation or torpor. Two large animal hibernators (Eurasian Brown Bear, American Black Bear) represent large animal hibernators that typically undergo a prolonged hibernation. Two small animal hibernators (Garden Dormouse, 13 Lined Ground Squirrel) undergo torpor with more substantial reductions in heart rate and body temperature, but whose torpor bouts are interrupted by short arousals that bring the animals back to near-summer like metabolic conditions.

Especially interesting, the investigators analyze the impact that body temperature may have on myosin ATP utilization by performing assays at two different temperatures (8 and 20 degrees C, in 13 Lined Ground Squirrels).

The multiple assays utilized provide a more comprehensive set of methods with which to test their hypothesis that muscle myosins change their metabolic efficiency during hibernation.

Suggestions and potential Weaknesses:

The following highlight comments from the first Public Review that this reviewer acknowledges authors may not be able to address in the current study but may merit carrying to the revised article of record.

(1) Statistical Analysis

The revised manuscript addresses the substantial issues. The two remaining questions may be noted for future experimental design(s): 1.c. That myosin isoforms may be considered a main effect and 1.e. The importance of biological vs statistical significance, especially for the mant-ATP chase data from the American Black Bear, where there appear to be shifts between the summer and winter data.

(2). Consistency of DRX/SRX data.

The responses to the first Public Review on the prior version of this manuscript highlight that a potential disconnect between the mant-ATP-predicted SRX:DRX proportions and x-ray diffraction studies measuring the position of the myosin heads (Mohran et al PMID 38103642) may be outside of the scope of the current manuscript. The reviewer accepts that a substantial discussion is outside of this article, but considers a brief mention possible differences between ATP kinetics and structural movements of value.

Overall, the manuscript represents a valuable data set comparing myosin properties of skeletal muscles multiple species exhibiting different forms of hibernation/torpor.

---

## [Author Response]

The following is the authors’ response to the original reviews.

**Public Reviews:**

**Reviewer #1:**
Summary:The evolution of non-shivering thermogenesis is of fundamental importance to understand. Here, in small mammals, the contractile apparatus of the muscle is shown to increase energy expenditure upon a drop in ambient temperature. Additionally, in the state of torpor, small hibernators did not show an increase in energy expenditure under the same challenge.Strengths:The authors have conducted a very well-planned study that has sampled the muscles of large and small hibernators from two continents. Multiple approaches were then used to identify the state of the contractile apparatus, and its energy expenditure under torpor or otherwise.Weaknesses:There was only one site of biopsy from the animals used (leg). It would be interesting to know if non-shivering thermogenesis is something that is regionally different in the animal, given the core body and distal limbs have different temperatures.

We thank the reviewer for their time and effort in reviewing our manuscript. Furthermore, we agree that it would be of interest to perform similar experiments upon different muscle sites in these animals. This is of particular interest as in some mammals, such as mice, distal limbs do not shiver and therefore non-shivering thermogenesis may play a more prominent role in heat regulation. A paper from Aydin et al., demonstrated that when shivering muscles (soleus) were prevented undergoing non-shivering thermogenesis via knock-out of UCP1 and were then exposed to cold temperatures, the force production of these muscles was significantly reduced due to prolonged shivering [1]. These results do suggest that even in shivering muscle, non-shivering thermogenesis plays a key role in the generation of heat for survival and for the maintenance of muscle performance. Furthermore, there is evidence from garden dormice that muscle temperature during torpor is slightly warmer than abdominal temperature and slighter cooler that heart temperature which is 7-8°C than abdominal suggesting the existence of non-shivering thermogenesis in skeletal and cardiac muscles (Giroud et al. in prep) [2]. We have added this information and reference into our discussion to reflect this important point (Discussion, paragraph 6, “As the biopsies which were used…”).

**Reviewer #2:**
Summary:The authors utilized (permeabilized) fibers from muscle samples obtained from brown and black bears, squirrels, and Garden dormice, to provide interesting and valuable data regarding changes in myosin conformational states and energetics during hibernation and different types of activity in summer and winter. Assuming that myosin structure is similar between species then its role as a regulator of metabolism would be similar and not different, yet the data reveal some interesting and perplexing differences between the selected hibernating species.Strengths:The experiments on the permeabilized fibers are complementary, sophisticated, and well-performed, providing new information regarding the characteristics of skeletal muscle fibers between selected hibernating mammalian species under different conditions (summer, interarousal, and winter).The studies involve complementary assessments of muscle fiber biochemistry, sarcomeric structure using X-ray diffraction, and proteomic analyses of posttranslational modifications.Weaknesses:It would be helpful to put these findings on permeabilized fibers into context with the other anatomical/metabolic differences between the species to determine the relative contribution of myosin energetics (with these other contributors) to overall metabolism in these different species, including factors such as fat volume/distribution.

We thank the reviewer for the time and effort they have put into reviewing our paper and are grateful for the helpful suggestions which we believe, enhances our work (please see below for detailed answers to critics).

**Reviewer #3:**
Summary and strengths:The manuscript, "Remodelling of skeletal muscle myosin metabolic states in hibernating mammals", by Lewis et al, investigates whether myosin ATP activity may differ between states of hibernation and activity in both large and small mammals. The study interrogates (primarily) permeabilized muscle strips or myofibrils using several state-of-the-art assays, including the mant-ATP assay to investigate ATP utilization of myosin, X-ray diffraction of muscles, proteomics studies, metabolic tests, and computational simulations. The overall data suggests that ATP utilization of myosin during hibernation is different than in active conditions.A clear strength of this study is the use of multiple animals that utilize two different states of hibernation or torpor. Two large animal hibernators (Eurasian Brown Bear, American Black Bear) represent large animal hibernators that typically undergo prolonged hibernation. Two small animal hibernators (Garden Dormouse, 13 Lined Ground Squirrel) undergo torpor with more substantial reductions in heart rate and body temperature, but whose torpor bouts are interrupted by short arousals that bring the animals back to near-summer-like metabolic conditions.Especially interesting, the investigators analyze the impact that body temperature may have on myosin ATP utilization by performing assays at two different temperatures (8 and 20 degrees C, in 13 Lined Ground Squirrels).The multiple assays utilized provide a more comprehensive set of methods with which to test their hypothesis that muscle myosins change their metabolic efficiency during hibernation.

We thank this reviewer for the effort and time they have put into carefully reviewing our manuscript and have taken on board their valuable suggestions to improve our manuscript (please see below for detailed answers to critics).

Suggestions and potential weaknesses:While the samples and assays provide a robust and comprehensive coverage of metabolic needs and testing, the data is less categorical. Some of these may be dependent on sample size or statistical analysis while others may be dependent on interpretation.(1) Statistical Analysis(1a) The results of this study often cannot be assessed properly due to a lack of clarity in the statistical tests.For example, the results related to the large animal hibernators (Figure 1) do not describe the statistical test (in the text of the results, methods, or figure legends). (Similarly for figure 6 and Supplemental Figure 1). Further, it is not clear whether or when the analysis was performed with paired samples. As the methods described, it appears that the Eurasian Brown Bear data should be paired per animal.

We thank the reviewer for these important points and have added information upon the statistical tests used where previously missing in each figure legend. Details on the statistical testing used for figure 6 are listed in the methods section, paragraph 18, “All statistical analysis of TMT derived protein expression data…”

(1b) The statistical methods state that non-parametric testing was utilized "where data was unevenly distributed". Please clarify when this was used.

We have now clariid all statistical tests used in the figure legends.

(1c) While there are two different myosin isoforms, the isoform may be considered a factor. It is unclear why a one-way ANOVA is generally used for most of the mant-ATP chase data.

The reviewer is right, in our analysis, we haven’t considered ‘myosin isoforms’ as a factor. One of the main reasons for that is because we have decided to treat fibres expressing different myosin heavy chain isoforms as totally separated entities (not interconnected).

(1d) While the technical replicates on studies such as the mant-ATP chase assay are well done, the total biological replicates are small. A consideration of the sample power should be included.

Unfortunately, obtaining additional biological samples from these unique species is challenging. Hence, we have added a statement in the Discussion section. This statement focuses on the potential benefits of increasing sample size to increase statistical power Discussion, paragraph 2, “In contrast to our study hypothesis…”

(1e) An analysis of the biological vs statistical significance should be considered, especially for the mant-ATP chase data from the American Black Bear, where there appear to be shifts between the summer and winter data.

We agree that it is important to be careful when drawing conclusions from data only based on p-values. We agree that the modest differences observed in these data on American Black bear, whilst not significant, are worth noting and we have added these considerations into the manuscript (Discussion, paragraph 2, “In contrast to our study hypothesis…).

(2) Consistency of DRX/SRX data.(2a) The investigators performed both mant-ATP chase and x-ray diffraction studies to investigate whether myosin heads are in an "on" or "off" state. The results of these two studies do not appear to be fully consistent with each other, which should not be a surprise. The recent work of Mohran et al (PMID 38103642) suggests that the mant-ATP-predicted SRX:DRX proportions are inconsistent with the position of the myosin heads. The discussion appears to lack a detailed assessment of this prior work and lack a substantive assessment contrasting the differing results of the two assays in the current study. i.e. why the current study's mant-ATP chase and x-ray diffraction results differ.

Prior works on skeletal muscle (observing discrepancies between Mant-ATP chase assay and X-ray diffraction) are rather scarce. Adding a comprehensive discussion about this may be beyond the scope of current study and would distract the reader from the main topic. For this reason, we have not added any section. Note that, we have other manuscripts in preparation that are specifically dedicated to the discrepancy.

(2b) The discussion of the current study's x-ray diffraction data relating to the I_1,1/I_1,0 ratio and how substantially different this is to the M6 results merits discussion. i.e. how can myosin both be more primed to contract during IBA versus torpor (according to intensity ratio), but also have less mass near the thick filament (M6).

The I1,1/I1,0 ratio indicates a subtle mass shift towards the myosin thick filament whilst the M6 spacing shows a more compliant thick filament. These results are not incompatible and rely on interpretation of the X-ray diffraction patterns. To avoid any confusion and avoid distracting the reader from the main topic, we have decided not to speculate there.

(3) Possible interactions with Heat Shock ProteinsHeat Shock Proteins (HSPs), such as HSP70, have been shown to be differential during torpor vs active states. A brief search of HSP and myosin reveals HPSs related to thick filament assembly and Heat Shock Cognate 70 interacting with myosin binding protein C. Especially given the author's discussion of protein stability and the potential interaction with myosin binding protein C and the SRX state, the limitation of not assessing HSPs should be discussed. (While HSP's relation to thick filament assembly might conceivably modify the interpretation of the M3 x-ray diffraction results, this reviewer acknowledges that possibility as a leap.)

The reviewer raises an interesting and potentially important of the potential impact of HSP and their interaction with the thick filament during hibernation. We have added a section into the discussion of this manuscript regarding this, with particular impact upon the HSP70 acting as a chaperone for myosin binding protein, however we feel that it is important to point out that HSPs have only been shown to interact with MYBPC3, a cardiac isoform of this protein which is not present in skeletal muscle [3]. (Discussion, paragraph 5, “Of potential further interest to the regulation of myosin…”).

Despite the above substantial concerns/weaknesses, this reviewer believes that this manuscript represents a valuable data set.Other comments related to interpretation:(4) The authors briefly mention the study by Toepfer et al [Ref 25] and that it utilizes cardiac muscles. There would benefit from increased discussion regarding the possible differences in energetics between cardiac and skeletal muscle in these states.

As this manuscript focuses solely on skeletal muscle. We believe that introducing comparisons between cardiac and skeletal muscles would confuse the reader. These types of muscles have very different regulations of SRX/DRX as an example. Note that we are preparing a manuscript focusing on cardiac muscle and hibernation.

(5) The author's analysis of temperature is somewhat limited.(5a) First, the authors use 20 degrees C (room temperature), not 37 degrees C, a more physiologic body temperature for large mammals. While it is true that limbs are likely at a lower temperature, 20 degrees C seems substantially outside of a normal range. Thus, temperature differences may have been minimized by the author's protocol.

The authors agree that the experimental set up to perform these single fiber studies at slightly higher temperatures may have been more beneficial to replicate the physiological conditions of these hind leg muscle in the analyzed animals. However, previous work has shown that the resting myosin dynamics are in fact stable at temperatures between 20-30 degrees Celsius in type I, type II and cardiac mammalian muscle fibers [4].

(5b) Second, the authors discuss the possibility of myosin contributing to non-shivering thermogenesis. The magnitude of this impact should be discussed. The suggestion of myosin ATP utilization also implies that there is some basal muscle tone (contraction), as the myosin ATPase utilizes ATP to release from actin, before binding and hydrolyzing again. Evidence of this tone should be discussed.

The reviewer is raising an interesting point and it would indeed be interesting to assess the magnitude of the impact and whether a basal muscle tone exists. Assessing the magnitude of the impact, is not an easy task and would require very advanced simulations which we are not experts in unfortunately. As for basal muscle tone, this is difficult to say as myosin is not actually binding to actin but hydrolyzing ATP at a faster pace during hibernation. We then think that the relation between our data and basal muscle tone is unclear. Hence, we have decided not to discuss these points in the manuscript.

**Recommendations for the authors:**

**Reviewer #1 (Recommendations For The Authors):**
This is a very interesting paper. I have some minor suggestions to help improve it.Is there any way to estimate the contribution of contractile apparatus to energy expenditure in reference to what is being generated at SERCA in the resting muscle under the various states examined?

This is an interesting idea however, as far as we know, this would be challenging experimentally (in the hibernating mammals) and difficult to achieve in a reliable manner.

It is important to emphasize that while BAT has been traditionally seen to be the site of NST, the skeletal muscle is very important, especially in large mammals, where BAT is going to be a very small % of the body and unlikely to be able to adequately provide heat. The addition of the contractile apparatus to SERCA as a heat generator at rest is very important -- also, the activation of ryanodine receptor Ca2+ to increase the local [Ca2+] at SERCA to generate heat has also recently been shown and should be mentioned (Meizoso-Huesca et al 2022, PNAS; Singh et al 2023, PNAS) alongside the work of Bal et al 2012 etc...

We have included these mechanisms and references into the manuscript discussion [5, 6]. Discussion, paragraph 4, “A critical difference between the large hibernators…”

Are you able to report the likely proportion of type II fibers in the muscles you have sampled?

The fiber type breakdown for all animals used in this study is reported in supplementary table 1.

The sampling of muscle from the legs of live animals is sensible and convenient. Is it possible different muscles in the body have different levels of NST, changes in energy expenditure in torpor, and other states?

As discussed in the public review we have added to the discussion of this manuscript to reflect upon this important point of potentially different results from different muscle sites in these animals.

**Reviewer #2 (Recommendations For The Authors):**
Is it likely that the proportion of fast and slow myosin-heavy chains within the selected sample of myofibers from the different mammals contributes to the overall differences in the energetics of different conformational states? In living animals, how does the relative contribution of the energetics from different muscle fiber types compare with the contribution from other organs to the overall regulation of metabolism during activities in summer, winter, or periods of intermittent arousal?

Fiber types in mammals can be vastly different between species as well as having a considerable amount of plasticity to change within each species upon specific stimuli. Furthermore, some mammals also have specific myosin heavy chain isoforms which have considerable expression, for example, myosin heavy chain 2B which is expressed in rodents such as mice but not larger mammals such as humans.

In the manuscript, we demonstrate that there is no significant change in the ATP usage by myosin in resting muscle in any of the species which we examined (Fig 1 F, L; Fig 2 E, J). The relatively high mitochondrial density of type I fibers when compared to type II fibers may contribute to a higher overall requirement of energy storage primarily via lipid oxidation. However, mitochondrial respiration is heavily suppressed during hibernation, so questions remain over the overall energy demand in hibernating muscle beyond myosin [7]. The fact that myosin ATP demand is relatively preserved in hibernating muscle suggests that skeletal muscle may be a relatively energy-demanding organ even during hibernation, we speculate in the manuscript this may be due to the requirement of maintaining muscular tone and function during this period of prolonged immobilization. This may be of relevance when one considers the almost complete shutdown of organs involved with food intake and breakdown such as the stomach and liver during hibernation. Furthermore, heart rate and breathing rates are vastly suppressed. Altogether, whilst is it difficult at this point to make an accurate estimate of energy demands between the different organs of hibernators, our data points to skeletal muscle to be a relatively high energy demand organ during these periods. When considering the difference between fiber type, again our data suggests that both type I and type II fibers have relatively similar energy demands during hibernation.

The supplementary data are quite revealing as to how the myosin isoform composition is stable in some species but highly plastic in others in response to the same environmental/metabolic challenges. Why is the myosin heavy chain isoform (I and II) composition stable for brown bears but not for black bears between summer and winter? This is very interesting. For the Ground squirrel, there is remarkable plasticity between myosin heavy chain isoforms (I and II) between summer, interbout arousal, and torpor. Yet in the Garden Dormouse, the myosin heavy chain isoform (I and II) composition is stable between these three activity states. The inconsistencies between and within species are perplexing and worthy of closer interrogation.The measurements and role of myosin energetics in different conformational states are interesting but need to be explained in context with other metabolic regulators for these hibernating mammals, especially because some species show remarkable plasticity whereas others show remarkable stability. For example, compare brown and black bears which show differences in the response of myosin composition the activity, interbout arousal, and torpor. Ground squirrels show remarkable plasticity in myosin isoform composition between activity states (and likely metabolic differences), but the Garden Dormouse has a remarkably stable myosin isoform composition during the three metabolic/environmental challenges. What mechanisms facilitate these modifications in some but not other mammals, even those of similar size? The differences are very interesting, worthy of follow-up, and may well contribute to further understanding the significance of the energetics of different myosin conformational states.

We agree that the changes seen between these species are very interesting and worthy of further investigation. What would be of further interest would be to look at methods which would allow for even deeper phenotyping, such as single fiber proteomics, to allow for the assessment of the percentage of hybrid fibers and fibers undergoing any fiber type switch during hibernating periods. Our results do observe a modest, albeit not significant, increase in the number of type I muscle fibers in 13-lined ground squirrels and Garden dormice during torpor which is consistent with previous studies[8]. Previous studies have demonstrated that lower temperatures may promote a shift towards more oxidative type I muscle fibers in mammals[9]. This could be an explanation for why we see this specifically in the smaller hibernators, however as we demonstrate and discuss, these lower temperatures are vital for the survival of these smaller mammals during hibernation so it would be inconsistent to hypothesize that these shifts are for heat-production purposes. Further studies are warranted to understand the relevance of these shifts further, particularly those with a higher sample size. It would also be on interest to examine fiber type percentages during the progression these long hibernating periods to observe if these changes are progressive.

As for the triggers and mechanisms which facilitate these changes to myosin dynamics, this is of current investigation by the field. One which may be of particular relevance to the changes seen during hibernation would that of steroid hormones previous research has demonstrated that steroid hormone levels in make and female bears change differentially[10]. This may be of relevance as the steroid hormone estradiol has been shown to slow the resting myosin ATP turnover via the binding of myosin RLC[11]. Considering these studies, future work which looks at hibernating animals of each sex as different groups may be fruitful.

**Reviewer #3 (Recommendations For The Authors):**
i. PDF Pg 8- Results- 'Myosin temperature sensitivity is lost in relaxed skeletal muscles fibers of hibernating Ictidomys tridecemlineatus.': An extra comma appears to be placed between "temperature, decrease".ii. PDF Pg 9- Results- 'Hyper-phosphorylation of Myh2 predictably stabilizes myosin backbone in hibernating Ictidomys tridecemlineatus.' (last paragraph): A parenthesis needs to be closed upon the first reference to "supplemental figures 2 and 3".iii. PDF Pg 15- Methods- 'Samples collection and cryo-preservation'- The authors use the term "individuals" in the 2nd line. Consider using "subjects".iv. PDF Pg 15- Methods- 'Samples collection and cryo-preservation' (2nd paragraph)- define "subadult" in approximate months or years.v. PDF Pg 15- Methods- 'Samples collection and cryo-preservation' (2nd paragraph)- The authors state that brown bears were located in "February and again ... in late June". Was this order of operations always held? If so, a comment about how the potential ageing from the hibernation (especially if sub-adult transitions to adulthood in this period) should be included.

All samples were collected during the subadult period of the lifespan of each bear and therefore we do not think that there would be a potential aging affect observed considering the lifespan of this species to be 20-30 years.

vi. PDF Pg 15- Methods- 'Samples collection and cryo-preservation' (3rd paragraph)- The justification for deprivation of feeding of black bears 24 hours prior to euthanasia should be included. A comment on how this might impact post-translational modifications or gene expression should be included.

Animals are starved prior to prevent aspiration during euthanasia. Considering these samples are to be compared to animals which have not consumed food or water for five months the impact relative impact on PTMs and gene expression would be considered negligible.

vii. PDF Pg 17- Methods- 'Mant-ATP chase experiments' (just after normalized fluorescence equation): The "Where" may be lowercase.viii. PDF Pg 17- Methods- 'Mant-ATP chase experiments' (last paragraph): The protocol for myosin staining, along with the antibody identification (source, catalog number) should be included.ix. PDF Pg 18- Methods- 'Post-translational Modification Peptide mapping': Define the makeup of the acrylamide gel and/or the source and catalog number.x. PDF Pg 18- Methods- 'Post-translational Modification Peptide mapping': The authors state that "Gel bands were washed..." Please specify which protein bands and if multiple bands (i.e. multiple isoforms) were isolated.

We thank this reviewer for their careful reading of our manuscript, we have made the changes above as relevant.

Reference list

(1) Aydin, J., et al., Nonshivering thermogenesis protects against defective calcium handling in muscle. Faseb j, 2008. 22(11): p. 3919-24.

(2) Stickler, S., Regional body temperatures and fatty acid compositions in hibernating garden dormice: a focus on cardiac adaptions. 2022, Vienna: Vienna. p. v, 49 Seiten, Illustrationen.

(3) Glazier, A.A., et al., HSC70 is a chaperone for wild-type and mutant cardiac myosin binding protein C. JCI Insight, 2018. 3(11).

(4) Walklate, J., et al., Exploring the super-relaxed state of myosin in myofibrils from fast-twitch, slow-twitch, and cardiac muscle. Journal of Biological Chemistry, 2022. 298(3).

(5) Meizoso-Huesca, A., et al., Ca^2+^ leak through ryanodine receptor 1 regulates thermogenesis in resting skeletal muscle. Proceedings of the National Academy of Sciences, 2022. 119(4): p. e2119203119.

(6) Singh, D.P., et al., Evolutionary isolation of ryanodine receptor isoform 1 for muscle-based thermogenesis in mammals. Proceedings of the National Academy of Sciences, 2023. 120(4): p. e2117503120.

(7) Staples, J.F., K.E. Mathers, and B.M. Duffy, Mitochondrial Metabolism in Hibernation: Regulation and Implications. Physiology, 2022. 37(5): p. 260-271.

(8) Xu, R., et al., Hibernating squirrel muscle activates the endurance exercise pathway despite prolonged immobilization. Exp Neurol, 2013. 247: p. 392-401.

(9) Yu, J., et al., Effects of Cold Exposure on Performance and Skeletal Muscle Fiber in Weaned Piglets. Animals (Basel), 2021. 11(7).

(10) Frøbert, A.M., et al., Differential Changes in Circulating Steroid Hormones in Hibernating Brown Bears: Preliminary Conclusions and Caveats. Physiol Biochem Zool, 2022. 95(5): p. 365-378.

(11) Colson, B.A., et al., The myosin super-relaxed state is disrupted by estradiol deficiency. Biochemical and biophysical research communications, 2015. 456(1): p. 151-155.